# Outlier-Robust Gromov-Wasserstein for Graph Data

**Lemin Kong**
CUHK
lkong@se.cuhk.edu.hk

**Jiajin Li**
Stanford University
jiajinli@stanford.edu

**Jianheng Tang**
HKUST
jtangbf@connect.ust.hk

**Anthony Man-Cho So**
CUHK
manchoso@se.cuhk.edu.hk

## Abstract

Gromov-Wasserstein (GW) distance is a powerful tool for comparing and aligning probability distributions supported on different metric spaces. Recently, GW has become the main modeling technique for aligning heterogeneous data for a wide range of graph learning tasks. However, the GW distance is known to be highly sensitive to outliers, which can result in large inaccuracies if the outliers are given the same weight as other samples in the objective function. To mitigate this issue, we introduce a new and robust version of the GW distance called RGW. RGW features optimistically perturbed marginal constraints within a Kullback-Leibler divergence-based ambiguity set. To make the benefits of RGW more accessible in practice, we develop a computationally efficient and theoretically provable procedure using Bregman proximal alternating linearized minimization algorithm. Through extensive experimentation, we validate our theoretical results and demonstrate the effectiveness of RGW on real-world graph learning tasks, such as subgraph matching and partial shape correspondence.

## 1  Introduction

Gromov-Wasserstein distance (GW) [26, 28] acts as a main model tool in data science to compare data distributions on unaligned metric spaces. Recently, it has received much attention across a host of applications in data analysis, e.g., shape correspondence [24, 31, 36, 27], graph alignment and partition [37, 38, 15, 18, 14, 44], graph embedding and classification [41, 43], unsupervised word embedding and translation [3, 19], generative modeling across incomparable spaces [8, 45].

In practice, the robustness of GW distance suffers heavily from its sensitivity to outliers. Here, outliers mean the samples with large noise, which usually are far away from the clean samples or have different structures from the clean samples. The hard constraints on the marginals in the Gromov-Wasserstein distance require all the mass in the source distribution to be entirely transported to the target distribution, making it highly sensitive to outliers. When the outliers are weighted similarly as other clean samples, even a small fraction of outliers corrupted can largely impact the GW distance value and the optimal coupling, which is unsatisfactory in real-world applications.

To overcome the above issue, some recent works are trying to relax the marginal constraints of GW distance. [33] introduces a $L^1$ relaxation of mass conservation of the GW distance. However, this reformulation replaces the strict marginal constraint that the transport plan should be a joint distribution with marginals as specific distributions by the constraint that only requires the transport plan to be a joint distribution, which can easily lead to over-relaxation. On another front, [10] propose a so-called partial GW distance (PGW), which only transports a fraction of mass from source distribution to target distribution. The formulation of PGW is limited to facilitating mass

37th Conference on Neural Information Processing Systems (NeurIPS 2023).

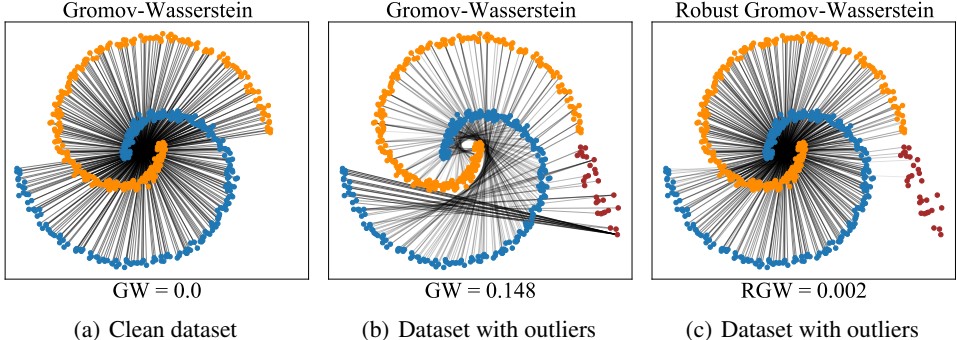

| Gromov-Wasserstein | Gromov-Wasserstein | Robust Gromov-Wasserstein |
|:---:|:---:|:---:|
| GW = 0.0 | GW = 0.148 | RGW = 0.002 |
| (a) Clean dataset | (b) Dataset with outliers | (c) Dataset with outliers |

Figure 1: Visualization of Gromov-Wasserstein couplings between two shapes, with the source in blue and the target in orange. In (a), the GW coupling without outliers is shown. In (b), the coupling with 10% outliers added to the target distribution is depicted. The sensitivity of GW to outliers is evident from the plot. In (c), we present the coupling generated by our proposed RGW formulation, which effectively disregards outliers and closely approximates the true GW distance.

destruction, which restricts its ability to handle situations where outliers exist predominantly on one side. A formulation that allows both mass destruction and creation is proposed in [35] called unbalanced GW (UGW). The UGW relaxes the marginal constraint via adding the quadratic $\varphi$-divergence as the penalty function in the objective function and extends GW distance to compare metric spaces equipped with arbitrary positive measures. Additionally, [40] proved that UGW is robust to outliers and can effectively remove the mass of outliers with high transportation costs. However, UGW is sensitive to the penalty parameter as it balances the reduction of outlier impact and the control of marginal distortion in the transport plan. On the computational side, an alternate Sinkhorn minimization method is proposed to calculate the entropy-regularized UGW. Note that the algorithm does not exactly solve UGW but approximates the lower bound of the entropic regularized UGW instead. From a statistical viewpoint, these works do not establish a direct link between the reformulated GW distance and the GW distance in terms of uncontaminated levels.

In this work, we propose the robust Gromov-Wasserstein (RGW) to estimate the GW distance robustly when dealing with outliers. To achieve this, RGW simultaneously optimizes the transport plan and selects the best marginal distribution from a neighborhood of the given marginal distributions, avoiding contaminated distributions. Perturbed marginal distributions help to re-weight the samples and lower the weight assigned to the outliers. The introduction of relaxed distributions to handle outliers draws inspiration from robust OT techniques [4, 29, 23]. Unlike robust OT, which is convex, RGW is non-convex, posing algorithmic challenges. This idea is also closely related to optimistic modelings of distribution ambiguity in data-driven optimization, e.g., upper confidence bound in the multi-armed bandit problem and reinforcement learning [7, 30, 1], data-driven distributionally robust decision-making with outliers [21, 9], etc.

Moreover, inspired by UGW, RGW relaxes the marginal constraint via adding the Kullback-Leibler (KL) divergence between the marginals of the transport plan and the perturbed distributions as the penalty function in the objective function to lessen the impact of the outliers further. Instead of utilizing the quadratic KL divergence as employed in unbalanced GW, we opt for KL divergence due to its computational advantages. It allows for convex subproblems with closed-form solutions, as opposed to the linearization required for non-convex quadratic KL divergence, which could be challenging algorithmically. Furthermore, we leverage the convexity of KL divergence to establish the statistical properties of RGW, leading to an upper bound by the true GW distance with explicit control through marginal relaxation parameters and marginal penalty parameters. This statistical advantage is disrupted by the non-convex nature of quadratic KL divergence. Additionally, KL divergence aligns with our goal of outlier elimination and is less sensitive to outliers compared to quadratic KL divergence, which is more outlier-sensitive due to its quadratic nature. Overall, RGW combines the introduction of perturbed marginal distributions with the relaxation of hard marginal constraints to achieve greater flexibility, allowing control over marginal distortion through marginal penalty parameters and reduction of outlier impact using marginal relaxation parameters.

To realize the modeling benefits of RGW, we further propose an efficient algorithm based on the Bregman proximal alternating linearized minimization (BPALM) method. The updates in each

iteration of BPALM can be computed in a highly efficient manner. On the theoretical side, we prove that the BPALM algorithm converges to a critical point of the RGW problem. Empirically, we demonstrate the effectiveness of RGW and the proposed BPALM algorithm through extensive numerical experiments on subgraph alignment and partial shape correspondence tasks. The results reveal that RGW surpasses both the balanced GW-based method and the reformulations of GW, including PGW and UGW.

**Our Contributions** We summarize our main contributions as follows:

- We develop a new robust model called RGW to alleviate the impact of outliers on the GW distance. The key insight is to simultaneously optimize the transport plan and perturb marginal distributions in the most efficient way.
- On the statistical side, we demonstrate that the robust Gromov-Wasserstein is bounded above by the true GW distance under the Huber $\epsilon$-contamination model.
- On the computational side, we propose an efficient algorithm for solving RGW using the BPALM method and prove that the algorithm converges to a critical point of the RGW problem.
- Empirical results on subgraph alignment and partial shape correspondence tasks demonstrate the effectiveness of RGW. This is the first successful attempt to apply GW-based methods to partial shape correspondence, a challenging problem pointed out in [36].

## 2 Problem Formulation

In this section, we review the definition of Gromov-Wasserstein distance and formally formulate the robust Gromov-Wasserstein. Following that, we discuss the statistical properties of the proposed robust Gromov-Wasserstein model under the Huber $\epsilon$-contamination model.

For the rest of the paper, we will use the following notation. Let $(X, d_X)$ be a complete separable metric space and denote the finite, positive Borel measure on $X$ by $\mathcal{M}_+(X)$. Let $\mathcal{P}(X) \subset \mathcal{M}_+(X)$ denotes the space of Borel probability measures on $X$. We use $\Delta^n$ to denote the simplex in $\mathbb{R}^n$. We use $\mathbf{1}_n$ and $\mathbf{1}_{n \times m}$ to denote the $n$-dimensional all-one vector and $n \times m$ all-one matrix. We use $\mathcal{S}^n$ to denote the set of $n \times n$ symmetric matrice. The indicator function of set $C$ is denoted as $\mathbb{I}_C(\cdot)$.

### 2.1 Robust Gromov-Wasserstein

The Gromov-Wasserstein (GW) distance aims at matching distributions defined in different metric spaces. It is defined as follows:

**Definition 2.1** (Gromov-Wasserstein). Suppose that we are given two unregistered complete separable metric spaces $(X, d_X)$, $(Y, d_Y)$ accompanied with Borel probability measures $\mu, \nu$ respectively. The GW distance between $\mu$ and $\nu$ is defined as

$$\inf_{\pi \in \Pi(\mu, \nu)} \iint |d_X(x, x') - d_Y(y, y')|^2 d\pi(x, y) d\pi(x', y'),$$

where $\Pi(\mu, \nu)$ is the set of all probability measures on $X \times Y$ with $\mu$ and $\nu$ as marginals.

As shown in the definition, the sensitivity to outliers of Gromov-Wasserstein distance is due to its hard constraints on marginal distributions. This suggests relaxing the marginal constraints such that the weight assigned to the outliers by the transport plan can be small. To do it, we invoke the Kullback-Leibler divergence, defined as $d_{\mathbf{KL}}(\alpha, \mu) = \int_X \alpha(x) \log\left(\alpha(x)/\mu(x)\right) dx$, to soften the constraint on marginal distributions. We also introduce an optimistically distributionally robust mechanism to perturb the marginal distributions and reduce the weight assigned to outliers. Further details on this mechanism will be discussed later.

**Definition 2.2** (Robust Gromov-Wasserstein). Suppose that we are given two unregistered complete separable metric spaces $(X, d_X)$, $(Y, d_Y)$ accompanied with Borel probability measures $\mu, \nu$ respectively. The Robust GW between $\mu$ and $\nu$ is defined as

$$\begin{aligned} \mathrm{GW}^{\mathrm{rob}}_{\rho_1, \rho_2}(\mu, \nu) := \min_{\alpha \in \mathcal{P}(X), \, \beta \in \mathcal{P}(Y)} \quad & F(\alpha, \beta) \\ \text{s.t. } d_{\mathbf{KL}}(\mu, \alpha) \leq \rho_1, \, & d_{\mathbf{KL}}(\nu, \beta) \leq \rho_2, \end{aligned} \tag{1}$$

where $F(\alpha, \beta) =$

$$\inf_{\pi \in \mathcal{M}^+(X \times Y)} \iint |d_X(x, x') - d_Y(y, y'))|^2 d\pi(x, y) d\pi(x', y') + \tau_1 d_{\mathbf{KL}}(\pi_1, \alpha) + \tau_2 d_{\mathbf{KL}}(\pi_2, \beta),$$

and $(\pi_1, \pi_2)$ are two marginals of the joint distribution $\pi$, defined by $\pi_1(A) = \pi(A \times Y)$ for any Borel set $A \subset X$ and $\pi_2(B) = \pi(X \times B)$ for any Borel set $B \subset Y$.

The main idea of our formulation is to optimize the transport plan and perturbed distribution variables in the ambiguity set of the observed marginal distributions jointly. This formulation aims to find the perturbed distributions that approximate the clean distributions and compute the transport plan based on the perturbed distributions. However, incorporating the constraints of equal marginals between the transport plan $\pi$ and the perturbed distributions $\alpha$ and $\beta$ directly poses challenges in developing an algorithm due to potential non-smoothness issues. Inspired by [35], we address this challenge by relaxing these marginal constraints and incorporating the KL divergence terms, denoted as $d_{\mathbf{KL}}(\pi_1, \alpha)$ and $d_{\mathbf{KL}}(\pi_2, \beta)$, into the objective function as penalty functions. Different from [35], we use KL divergence instead of quadratic KL divergence due to its joint convexity, which is more amenable to algorithm development, as quadratic KL divergence is typically non-convex. Besides, transforming the hard marginal constraints into penalty functions can further lessen the impact of outliers on the transport plan.

Our new formulation extends the balanced GW distance and can recover it by choosing $\rho_1 = \rho_2 = 0$ and letting $\tau_1$ and $\tau_2$ tend to infinity. When properly chosen, $\rho_1$ and $\rho_2$ can encompass the clean distributions within the ambiguity sets. In this scenario, the relaxed reformulation closely approximates the original GW distance in a certain manner. Building on this concept, we prove that RGW can serve as a robust approximation of the GW distance without outliers, given some mild assumptions on the outliers.

## 2.2 Robustness Guarantees

Robust Gromov-Wasserstein aims at mitigating the sensitivity of the GW distance to outliers, which can result in large inaccuracies if the outliers are given the same weight as other samples in the objective function. Specifically, RGW is designed to address the issue of the GW distance exploding as the distance between the clean samples and the outliers goes to infinity. In general, even a small number of outliers can cause the GW distance to change dramatically when added to the marginal distributions. To formalize this, consider the Huber $\epsilon$-contamination model popularized in robust statistics [22, 11, 12]. In that model, a base measure $\mu_c$ is contaminated by an outlier distribution $\mu_a$ to obtain a contaminated measure $\mu$,

$$\mu = (1 - \epsilon)\mu_c + \epsilon\mu_a. \tag{2}$$

Under this model, data are drawn from $\mu$ defined in (2).

Under the assumption of the Huber $\epsilon$-contamination model, it can be demonstrated that by selecting suitable values of $\rho_1$ and $\rho_2$, the robust Gromov-Wasserstein distance ensures that outliers are unable to substantially inflate the transportation distance. For robust Gromov-Wasserstein, we have the following bound:

**Theorem 2.3.** *Let $\mu$ and $\nu$ be two distributions corrupted by fractions $\epsilon_1$ and $\epsilon_2$ of outliers, respectively. Specifically, $\mu$ is defined as $(1 - \epsilon_1)\mu_c + \epsilon_1\mu_a$, and $\nu$ is defined as $(1 - \epsilon_2)\nu_c + \epsilon_2\nu_a$, where $\mu_c$ and $\nu_c$ represent the clean distributions, and $\mu_a$ and $\nu_a$ represent the outlier distributions. Then,*

$$\mathrm{GW}_{\rho_1, \rho_2}^{\mathrm{rob}}(\mu, \nu) \leq \mathrm{GW}(\mu_c, \nu_c) + \max\left(0, \epsilon_1 - \frac{\rho_1}{d_{\mathbf{KL}}(\mu_a, \mu_c)}\right) \tau_1 d_{\mathbf{KL}}(\mu_c, \mu_a)$$

$$+ \max\left(0, \epsilon_2 - \frac{\rho_2}{d_{\mathbf{KL}}(\nu_a, \nu_c)}\right) \tau_2 d_{\mathbf{KL}}(\nu_c, \nu_a).$$

In Appendix B, we provide a proof that constructs a feasible transport plan and relaxed marginal distributions. By relaxing the strict marginal constraints, we can find a feasible transport plan that closely approximates the transport plan between the clean distributions and obtain relaxed marginal distributions that approximate the clean distributions.

The derived bound indicates that robust Gromov-Wasserstein provides a provably robust estimate under the Huber $\epsilon$-contamination model. If the fraction of outliers is known, the upper bound for the robust GW is determined by the true Gromov-Wasserstein distance, along with additional terms that account for the KL divergence between the clean distribution and the outlier distribution for both $\mu$ and $\nu$. The impact of this factor is determined by the extent of relaxation in the marginal distributions $\rho_1$ and $\rho_2$. By carefully choosing $\rho_1 = \epsilon_1 d_{\mathbf{KL}}(\mu_a, \mu_c)$ and $\rho_2 = \epsilon_2 d_{\mathbf{KL}}(\nu_a, \nu_c)$, we can tighten the upper bound on the robust GW value, while still keeping it below the original GW distance (excluding outliers). Importantly, substituting these values of $\rho_1$ and $\rho_2$ yields $\mathrm{GW}_{\rho_1, \rho_2}^{\mathrm{rob}}(\mu, \nu) \leq \mathrm{GW}(\mu_c, \nu_c)$, indicating that the robust GW between the contaminated distribution $\mu$ and $\nu$ is upper bounded by the original GW distance between the clean distribution $\mu_c$ and $\nu_c$.

**Remark 2.4.** The following inequality for UGW under the Huber $\epsilon$-contamination model can be derived using the same techniques as in Theorem 2.3:

$$\mathrm{UGW}(\mu, \nu) \leq \mathrm{GW}(\mu_c, \nu_c) + \tau_1 d_{\mathbf{KL}}^{\otimes}(\mu_c, \mu) + \tau_2 d_{\mathbf{KL}}^{\otimes}(\nu_c, \nu).$$

We observe that the terms $d_{\mathbf{KL}}^{\otimes}(\mu_c, \mu)$ and $d_{\mathbf{KL}}^{\otimes}(\mu_c, \mu)$ cannot be canceled out unless $\tau_1$ and $\tau_2$ are set to zero, resulting in over-relaxation. However, RGW allows us to control the error terms $d_{\mathbf{KL}}(\mu_c, \mu_a)$ and $d_{\mathbf{KL}}(\nu_c, \nu_a)$ through the marginal relaxation parameters $\rho_1$ and $\rho_2$.

# 3 Proposed Algorithm

## 3.1 Problem Setup

To start with our algorithmic developments, we consider the discrete case for simplicity and practicality, where $\mu$ and $\nu$ are two empirical distributions, i.e., $\mu = \sum_{i=1}^{n} \mu_i \delta_{x_i}$ and $\nu = \sum_{j=1}^{m} \nu_j \delta_{y_j}$. Denote $D \in \mathcal{S}^n$, $D_{ik} = d_X(x_i, x_k)$ and $\bar{D} \in \mathcal{S}^m$ and $\bar{D}_{jl} = d_Y(y_j, y_l)$. We construct a 4-way tensor as follows:

$$\mathcal{L}(D, \bar{D}) := \left( |d_X(x_i, x_k) - d_Y(y_j, y_l)|^2 \right)_{i,j,k,l}.$$

We define the tensor-matrix multiplication as

$$(\mathcal{L} \otimes T)_{ij} := \left( \sum_{k,\ell} \mathcal{L}_{i,j,k,\ell} T_{k,\ell} \right)_{i,j}.$$

Then, the robust GW admits the following reformulation:

$$\begin{aligned}
\min_{\pi, \alpha, \beta} \quad & \langle \mathcal{L}(D, \bar{D}) \otimes \pi, \pi \rangle + \tau_1 d_{\mathbf{KL}}(\pi_1, \alpha) + \tau_2 d_{\mathbf{KL}}(\pi_2, \beta) \\
\text{s.t.} \quad & d_{\mathbf{KL}}(\mu, \alpha) \leq \rho_1, d_{\mathbf{KL}}(\nu, \beta) \leq \rho_2, \\
& \alpha \in \Delta^n, \beta \in \Delta^m, \pi \geq 0.
\end{aligned} \tag{3}$$

Here, $\pi_1 = \pi \mathbf{1}_m$ and $\pi_2 = \pi^T \mathbf{1}_n$.

## 3.2 Bregman Proximal Alternating Linearized Minimization (BPALM) Method

As Problem (3) is non-convex and involves three variables, we employ BPALM [5, 2] to solve it. By choosing the KL divergence as Bregman distance, the updates of this algorithm are given by:

$$\pi^{k+1} = \arg\min_{\pi \geq 0} \left\{ \langle \mathcal{L}(D, \bar{D}) \otimes \pi^k, \pi \rangle + \tau_1 d_{\mathbf{KL}}(\pi_1, \alpha^k) + \tau_2 d_{\mathbf{KL}}(\pi_2, \beta^k) + \frac{1}{t_k} d_{\mathbf{KL}}(\pi, \pi^k) \right\}, \tag{4}$$

$$\alpha^{k+1} = \arg\min_{\substack{\alpha \in \Delta^n \\ d_{\mathbf{KL}}(\mu, \alpha) \leq \rho_1}} \left\{ d_{\mathbf{KL}}(\pi_1^{k+1}, \alpha) + \frac{1}{c_k} d_{\mathbf{KL}}(\alpha^k, \alpha) \right\}, \tag{5}$$

$$\beta^{k+1} = \arg\min_{\substack{\beta \in \Delta^m \\ d_{\mathbf{KL}}(\nu, \beta) \leq \rho_2}} \left\{ d_{\mathbf{KL}}(\pi_2^{k+1}, \beta) + \frac{1}{r_k} d_{\mathbf{KL}}(\beta^k, \beta) \right\}. \tag{6}$$

Here, $t_k$, $c_k$, and $r_k$ are stepsizes in BPALM.

In our algorithm updates, we employ distinct proximal operators for $\pi$ and $\alpha$ (and $\beta$). The use of $d_{\mathbf{KL}}(\pi, \pi^k)$ in $\pi$-subproblem (4) allows for the application of the Sinkhorn algorithm, while the introduction of $d_{\mathbf{KL}}(\alpha^k, \alpha)$ in $\alpha$-subproblem (5) facilitates a closed-form solution, which we will detail in the following part.

To solve the $\pi$-subproblem, we can utilize the Sinkhorn algorithm for the entropic regularized unbalanced optimal transport problem. This algorithm, which has been previously introduced in [13, 32], is well-suited for our needs. As for the $\alpha$-subproblem, we consider the case where $\rho_1$ is strictly larger than 0. Otherwise, when $\rho_1 = 0$, $\alpha$ should simply equal $\mu$, making the subproblem unnecessary. To solve the $\alpha$-subproblem, we attempt to find the optimal dual multiplier $w^*$. Specifically, consider the problem:

$$\min_{\alpha \in \Delta^n} d_{\mathbf{KL}}(\pi_1^{k+1}, \alpha) + \frac{1}{c_k} d_{\mathbf{KL}}(\alpha^k, \alpha) + w(d_{\mathbf{KL}}(\mu, \alpha) - \rho_1). \tag{7}$$

Let $\hat{\alpha}(w)$ represent the optimal solution to (7), and we define the function $p : \mathbb{R}_+ \to \mathbb{R}$ by $p(w) = d_{\mathbf{KL}}(\mu, \hat{\alpha}(w)) - \rho_1$. We prove the convexity, differentiability, and monotonicity of $p$, which are crucial for developing an efficient algorithm for (5) later.

**Proposition 3.1.** *Problem* (7) *has a closed-form solution*

$$\hat{\alpha}(w) = \frac{\pi^{k+1} \mathbf{1}_m + \frac{1}{c_k} \alpha^k + w\mu}{\sum_{ij} \pi_{ij}^{k+1} + \frac{1}{c_k} + w}.$$

*If $w$ satisfies (i) $w = 0$ and $p(w) \leq 0$, or (ii) $w > 0$, $p(w) = 0$, then $\hat{\alpha}(w)$ is the optimal solution to the $\alpha$-subproblem* (5). *Moreover, $p(\cdot)$ is convex, twice differentiable, and monotonically non-increasing on $\mathbb{R}_+$.*

Given Proposition 3.1, we begin by verifying $p(0) \leq 0$. If this condition is not met, and given that $p(0) > 0$ while $\lim_{w \to +\infty} p(w) = -\rho_1 < 0$, it implies that $p$ possesses at least one root within $\mathbb{R}_+$. The following proposition provides the framework to seek the root of $p$ by employing Newton's method, with the initialization set at 0. Hence, the $\alpha$-subproblem can be cast to search a root of $p$ in one dimension, in which case it can be solved efficiently.

**Proposition 3.2.** *Let $p(\cdot) : I \to \mathbb{R}$ be a convex, twice differentiable, and monotonically non-increasing on the interval $I \subset \mathbb{R}$. Assume that there exists an $\tilde{x}, \bar{x} \in I$ such that $p(\tilde{x}) > 0$ and $p(\bar{x}) < 0$. Then $p$ has a unique root on $I$, and the sequence obtained from Newton's method with initial point $x_0 = \tilde{x}$ will converge to the root of $p$.*

Since the $\beta$-subproblem shares the same structure as the $\alpha$-subproblem, we can apply this method to search for the optimal solution to the $\beta$-subproblem.

## 3.3 Convergence Analysis

To illustrate the convergence result of BPALM, we consider the compact form for simplicity:

$$\min_{\alpha, \beta, \pi} F(\pi, \alpha, \beta) = f(\pi) + q(\pi) + g_1(\pi, \alpha) + g_2(\pi, \beta) + h_1(\alpha) + h_2(\beta),$$

where $f(\pi) = \langle \mathcal{L}(D, \bar{D}) \otimes \pi, \pi \rangle$, $q(\pi) = \mathbb{I}_{\{\pi \geq 0\}}(\pi)$, $g_1(\pi, \alpha) = \tau_1 d_{\mathbf{KL}}(\pi \mathbf{1}_m, \alpha)$, $g_2(\pi, \beta) = \tau_2 d_{\mathbf{KL}}(\pi^T \mathbf{1}_n, \beta)$, $h_1(\alpha) = \mathbb{I}_{\{\alpha \in \Delta^n, d_{\mathbf{KL}}(\mu, \alpha) \leq \rho_1\}}(\alpha)$, and $h_2(\beta) = \mathbb{I}_{\{\beta \in \Delta^m, d_{\mathbf{KL}}(\nu, \beta) \leq \rho_2\}}(\beta)$.

The following theorem states that any limit point of the sequence generated by BPALM belongs to the critical point set of problem (3).

**Theorem 3.3** (Subsequence Convergence). *Suppose that in Problem* (1), *the step size $t_k$ in* (4) *satisfies $0 < \underline{t} \leq t_k < \bar{t} \leq \sigma/L_f$ for $k \geq 0$ where $\underline{t}, \bar{t}$ are given constants and $L_f$ is the gradient Lipschitz constant of $f$. The step size $c_k$ in* (5) *and $r_k$ in* (6) *satisfy $0 < \underline{r} \leq c_k, r_k < \bar{r}$ for $k \geq 0$ where $\underline{r}, \bar{r}$ are given constants. Any limit point of the sequence of solutions $\{\pi^k, \alpha^k, \beta^k\}_{k \geq 0}$ belongs to the critical point set $\mathcal{X}$, where $\mathcal{X}$ is defined by*

$$\left\{ (\pi, \alpha, \beta) : \begin{array}{l} 0 \in f(\pi) + \partial q(\pi) + \nabla_\pi g_1(\pi, \alpha) + \nabla_\pi g_2(\pi, \beta), \\ 0 \in \nabla_\alpha g_1(\pi, \alpha) + \partial h_1(\alpha), \\ 0 \in \nabla_\beta g_2(\pi, \beta) + \partial h_2(\beta), \\ (\pi, \alpha, \beta) \in \mathbb{R}^{n \times m} \times \mathbb{R}^n \times \mathbb{R}^m \end{array} \right\}.$$

For the sake of brevity, we omit the proof. We refer the reader to Appendix C for further details.

## 4 Experiment Results

In this section, we present comprehensive experimental results to validate the effectiveness of our proposed RGW model and BPALM algorithm in various graph learning tasks, specifically subgraph alignment and partial shape correspondence. Traditionally, balanced GW has been applied successfully in scenarios where the source and target graphs have similar sizes. However, in our approach, we treat the missing part of the target graph as outliers and leverage RGW for improved performance. All simulations are conducted in Python 3.9 on a high-performance computing server running Ubuntu 20.10, equipped with an Intel(R) Xeon(R) Silver 4214R CPU. Our code is available at https://github.com/lmkong020/outlier-robust-GW.

### 4.1 Partial Shape Correspondence

In this subsection, we first investigate a toy matching problem in a 2D setting to support and validate our theoretical insights and results presented in Section 2. Figure 2 (a) illustrates an example where we aim to map a two-dimensional shape without symmetries to a rotated version of the same shape while accounting for outliers in the source domain. Here, we sample 300 points from the source shape and 400 points from the target shape. Additionally, we introduce 50 outliers by randomly adding points from a discrete uniform distribution on $[-3, -2.5] \times [0, 0.5]$ to the source domain. The distance matrices, $D$ and $\bar{D}$ are computed using pairwise Euclidean distances.

Figures 2 provide visualizations of the coupling matrices and objective values for all the models, highlighting the matching results. In Figure 2(c), it is evident that even a small number of outliers has a significant impact on the coupling and leads to an increased estimated GW distance. While unbalanced GW and partial GW attempt to handle outliers to some extent, they fall short in achieving accurate mappings. On the other hand, our robust GW formulation, RGW, effectively disregards outliers and achieves satisfactory performance. Additionally, the objective value of RGW closely approximates the true GW distance without outliers, as indicated by Theorem 2.3, approaching zero.

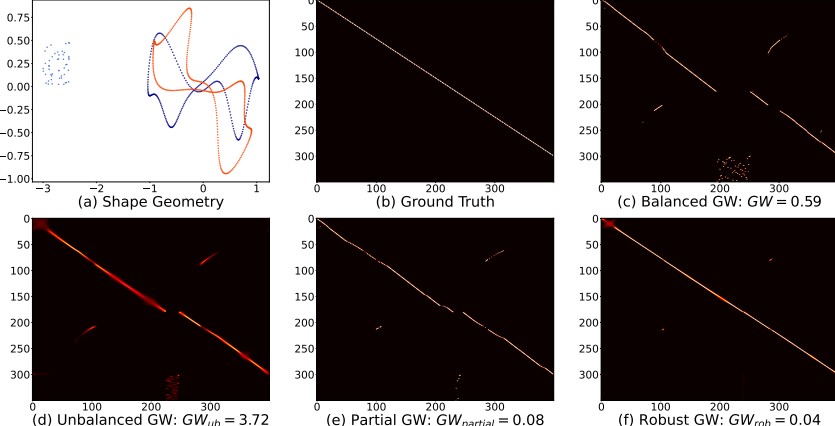

Figure 2: (a): 2D shape geometry of the source and target; (b)-(f): visualization of ground truth and the matching results of balanced GW, unbalanced GW, partial GW, and robust GW.

We evaluate the matching performance of RGW on the TOSCA dataset [6, 34] for partial shape correspondence. For the initial estimation of the transport plan in both RGW and PGW, we utilize the partial functional map method [34], employing 30 eigenfunctions. This method establishes an initial matching relationship between the source and target shapes. This method establishes an initial matching relationship between the source and target shapes by setting $\pi_{ik}$ to 1 for matching pairs $(i, k)$ and 0 otherwise. The resultant transport plan is scaled by $\|\pi\|_1 = \sum_{ij} \pi_{ij}$. UGW is not suitable for this large-scale task due to its long execution time. As shown in Figure 3, RGW outperforms PGW, and it enhances the performance of the initial point.

### 4.2 Subgraph Alignment

The subgraph alignment problem, which involves determining the isomorphism between a query graph and a subgraph of a larger target graph, has been extensively studied [16, 20]. While the

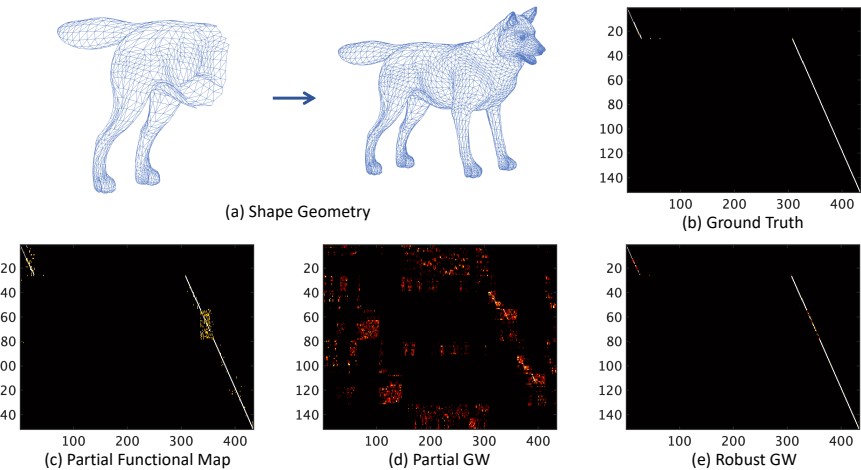

Figure 3: (a): 3D shape geometry of the source and target; (b)-(e): visualization of ground truth, initial point obtained from the partial functional map, and the matching results of PGW and RGW.

restricted quadratic assignment problem is commonly used for graphs of similar sizes, the GW distance provides an optimal probabilistic correspondence that preserves the isometric property. In the subgraph alignment context, the nodes in the target graph, excluding those in the source graph, can be considered outliers, making the RGW model applicable to this task. In our comparison, we evaluate RGW against various methods, including unbalanced GW, partial GW, semi-relaxed GW (srGW) [42], RGWD [25], and methods for computing balanced GW such as FW [39], BPG [46], SpecGW [15], eBPG [36], and BAPG [24].

**Database Statistics** We evaluate the methods on synthetic and real databases. In the synthetic database, we generate target graphs $\mathcal{G}_t$ using Barabasi-Albert models with scales ranging from 100 to 500 nodes. The source graphs $\mathcal{G}_s$ are obtained by sampling connected subgraphs of $\mathcal{G}_t$ with a specified percentage of nodes. This process results in five synthetic graph pairs for each setup, totaling 200 graph pairs. The *Proteins* and *Enzymes* biological graph databases from [15] are also used, following the same subgraph generation routine. For the *Proteins* database, we evaluate the accuracy of matching the overlap between two subgraphs with 90% overlap and between a subgraph and the entire graph, presented in the "Proteins-2" and "Proteins-1" columns, respectively. We compute the matching accuracy by comparing the predicted correspondence set $\mathcal{S}_{\text{pred}}$ to the ground truth correspondence set $\mathcal{S}_{\text{gt}}$, with accuracy calculated as Acc $= |\mathcal{S}_{\text{gt}} \cap \mathcal{S}_{\text{pred}}|/|\mathcal{S}_{\text{gt}}| \times 100\%$. In addition, we also evaluate our methods on the *Douban Online-Offline* social network dataset, which consists of online and offline graphs, representing user interactions and presence at social gatherings, respectively. The online graph includes all users from the offline graph, with 1,118 users serving as ground truth alignments. Node locations are used as features in both graphs. In line with previous works [18, 37], we gauge performance using the Hit@k metric, which calculates the percentage of nodes in set $\mathcal{V}_t$ where the ground truth alignment includes $\mathcal{V}_s$ among the top-k candidates.

Table 1: Comparison of the average matching accuracy (%) and wall-clock time (seconds) on subgraph alignment of 50% subgraph on datasets Synthetic, Proteins and Enzymes and Hit@1 and Hit@10 of dataset Douban.

| Method | Synthetic | | Proteins-1 | | Proteins-2 | | Enzymes | | Douban | |
|--------|------|------|------|------|------|------|------|------|-------|--------|
| | Acc | Time | Acc | Time | Acc | Time | Acc | Time | Hit@1 | Hit@10 |
| FW | 2.27 | 18.39 | 16.00 | 27.05 | 26.15 | 60.34 | 15.47 | 9.57 | 17.97 | 51.07 |
| SpecGW | 1.78 | 3.72 | 12.06 | 11.07 | 42.64 | 12.85 | 10.69 | 3.96 | 2.68 | 9.83 |
| eBPG | 3.71 | 85.31 | 19.88 | 1975.12 | 32.15 | 9645.05 | 21.58 | 1219.81 | 0.08 | 0.53 |
| BPG | 15.41 | 24.67 | 29.30 | 118.26 | 61.26 | 80.39 | 32.49 | 70.42 | 72.72 | 92.39 |
| BAPG | 48.89 | 27.95 | 30.98 | 122.13 | 66.84 | 16.49 | 35.64 | 16.41 | 72.18 | 92.58 |
| srGW | 1.60 | 152.01 | 21.30 | 63.00 | 12.08 | 172.48 | 24.13 | 19.68 | 4.03 | 11.54 |
| RGWD | 16.68 | 955.40 | 27.94 | 4396.41 | 59.69 | 3586.56 | 30.35 | 2629.00 | 4.11 | 16.46 |
| UGW | 89.88 | 176.24 | 25.72 | 4026.93 | 67.30 | 1853.82 | 43.73 | 1046.29 | 0.09 | 0.72 |
| PGW | 2.28 | 479.99 | 13.94 | 544.79 | 20.08 | 348.44 | 11.43 | 212.09 | 18.24 | 37.03 |
| RGW | **94.44** | 361.44 | **53.30** | 834.76 | **69.38** | 466.91 | **63.43** | 293.84 | **75.58** | **96.24** |

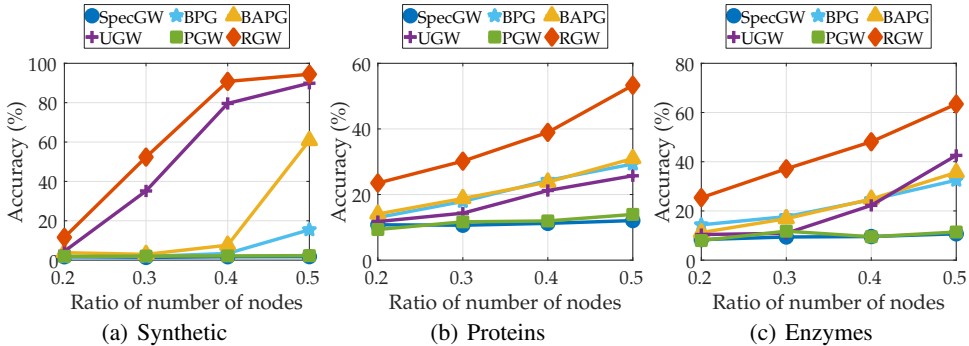

Figure 4: Graph Alignment performance for selected methods in relation to varying ratios of overlapping subgraphs on Synthetic, Protein, and Enzymes databases.

**Parameters Setup** We use unweighted symmetric adjacency matrices $D$ and $\bar{D}$ as input distance matrices. SpecGW employs the heat kernel $(\exp(-L))$ with the normalized graph Laplacian matrix $L$. Both $\mu$ and $\nu$ are set as uniform distributions. For SpecGW, BPG, eBPG, BAPG, srGW, and RGWD, we follow their respective paper setups. FW is implemented using the default PythonOT package. UGW selects the best results from regularization parameter sets $\{0.5, 0.2, 0.1, 0.01, 0.001\}$ and marginal penalty parameter sets $\{0.1, 0.01, 0.001\}$. PGW reports the best results in the range of transported mass from 0.1 to 0.9. RGW sets $\tau_1 = \tau_2 = 0.1$ and selects the best results from the ranges $\{0.05, 0.1, 0.2, 0.5\}$ for marginal relaxation parameters and $\{0.01, 0.05, 0.1, 0.5, 1\}$ for step size parameters $(t_k, c_k, r_k)$. The transport plan initialization uses different approaches depending on the dataset. For the *Synthetic*, *Proteins*, and *Enzymes* datasets, we employ the $\mathbf{1}_{n \times m}/(nm)$ approach. For the *Douban Online-Offline* dataset, we initialize the transport plan using classical optimal transport conducted on the feature space. Additionally, for all datasets, we initialize $\alpha$ and $\beta$ with uniform distributions.

**Result of All Methods** Table 1 and Figure 4 provide a comprehensive comparison of matching accuracy and computation time across various methods on the *Synthetic*, *Proteins*, and *Enzymes* datasets. Notably, RGW demonstrates superior accuracy compared to other methods, while maintaining comparable computation time with UGW and PGW. On the other hand, methods for computing the balanced GW exhibit poor performance on all datasets, particularly on the extensive *Synthetic* graph database. This can be attributed to their struggle in fulfilling the hard marginal constraint on the source side and addressing the presence of outliers. Furthermore, the introduction of local minima in the balanced GW problem due to partial source graph structures contributes to the suboptimal results. UGW performs well with low target graph partiality but suffers significant degradation as partiality increases. PGW, aimed at mitigating outlier impact, inadvertently affects the matching between clean samples by reducing the mass transported from the source domain. In addition, RGW achieved the best results with the highest Hit@1 and Hit@10 on the *Douban Online-Offline* dataset, improving performance from 4.04% and 14.9% respectively for the initial point created by features.

**Selection of Hyperparameters $\rho$ and $\tau$** The hyperparameters $\rho_1$, $\rho_2$, $\tau_1$, and $\tau_2$ in our formulation serve distinct roles. Specifically, $\rho_1$ and $\rho_2$ control the extent of marginal relaxation, while $\tau_1$ and $\tau_2$ act as the marginal penalty parameters. As indicated by Theorem 2.3, we have the flexibility to fix the value of $\tau$ and adjust $\rho$ to accommodate outlier effects. To determine optimal hyperparameter values, we conduct two analyses. First, we examine the effect of varying $\rho$ while maintaining a fixed subgraph node ratio and constant $\tau$ on the *Enzymes* dataset. Second, we adjust $\tau$ while holding $\rho$ constant and assess the impact across the *Synthesis*, *Proteins*, and *Enzymes* datasets in the task of matching a 50%

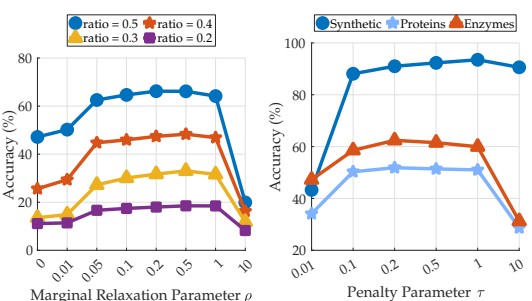

Figure 5: Sensitivity analysis of $\rho$ with fixed $\tau = 0.1$ on the Enzymes database, and sensitivity analysis of $\tau$ with fixed $\rho = 0.2$ on the Synthetic, Proteins, and Enzymes databases.

subgraph to the entire graph. In Figure 5, the left side demonstrates that while keeping the ratio and $\tau$ fixed but varying $\rho$, accuracy diminishes when $\rho$ becomes either excessively large or too small. However, within the range of 0.05 to 1, accuracy remains resilient, highlighting the mitigating effect of marginal relaxation on outlier influence. Notably, accuracy significantly improves when $\rho$ falls within the range of 0.05 to 1 compared to the scenario with $\rho = 0$, thus affirming the significance of marginal relaxation in our model. A similar trend is observed for $\tau$ on the right side of Figure 5, where maintaining a fixed ratio and $\rho$ while adjusting $\tau$ results in stable accuracy within the range of 0.1 to 1 but declining beyond this range. Consequently, in RGW computations, we can initially set $\rho$ to 0.2 and $\tau$ to 0.1 by default and subsequently adjust $\rho$ based on the outlier ratio: increasing it for a large ratio and decreasing it for a small ratio.

### 4.3 Tightness of the Bound in Theorem 2.3

Our primary focus is on scenarios where the GW distance between clean samples is nearly zero or zero due to noise, such as in partial shape correspondence and subgraph alignment tasks. In such cases, it is possible to find an isometric mapping from the query subgraph to a portion of the entire graph. By appropriately selecting the value of $\rho$, as discussed in Section 2.2, the upper bound in RGW becomes the GW distance between clean samples, which is zero. As RGW is always nonnegative, this upper bound is tight in this context.

To empirically validate this, we conducted experiments on the toy example in Section 4.1, and Figure 6 (a) illustrates the function values of PGW, UGW, and RGW with varying outlier ratios. The results confirm that the value of RGW can remain close to zero as the ratio of outliers increases. Additionally, Figure 6 (b) shows the function value of RGW and its upper bound as $\rho$ varies. Both the RGW value and its upper bound decrease, converging to zero as $\rho$ increases. This observation provides empirical support for Theorem 2.3. Regarding UGW and its upper bound with changing $\tau$, we observed that both the UGW value and its upper bound increase as $\tau$ becomes larger, as shown in Figure 6 (c). Unlike RGW, UGW's $\tau$ must strike a balance between reducing outlier impact and preserving marginal distortion in the transport plan. This demands a careful balance and caution against setting $\tau$ excessively close to zero, which could lead to over-relaxation and potentially deteriorate the performance.

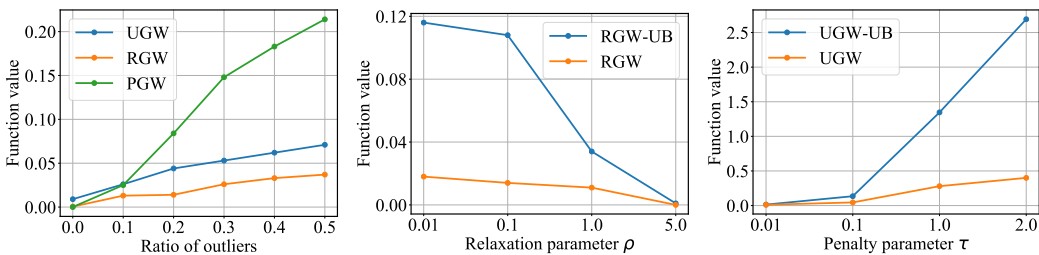

(a) Values of PGW, UGW, and RGW   (b) RGW value and upper bound   (c) UGW value and upper bound

Figure 6: (a) Function values of PGW, UGW, and RGW for varying $\epsilon$; (b) Function value of RGW and its upper bound for different $\rho$; (c) Function value of UGW and corresponding upper bound for different $\tau$.

## 5 Conclusion

In this paper, we introduce RGW, a robust reformulation of Gromov-Wasserstein that incorporates distributionally optimistic modeling. Our theoretical analysis demonstrates its robustness to outliers, establishing RGW as a reliable estimator. We propose a Bregman proximal alternating linearized minimization method to efficiently solve RGW. Extensive numerical experiments validate our theoretical results and demonstrate the effectiveness of the proposed algorithm. Regarding the robust estimation of Gromov-Wasserstein, a natural question is whether we can recover the transport plan from the RGW model. On the computational side, our algorithm suffers from the heavy computation cost due to the use of the unbalanced OT as our subroutine, which limits its application in large-scale real-world settings. To address this issue, a natural future direction is to develop single-loop algorithms to leverage the model benefits of robust GW for real applications.

## Acknowledgments and Disclosure of Funding

Anthony Man-Cho So is supported in part by the Hong Kong Research Grants Council (RGC) General Research Fund (GRF) project CUHK 14203920.

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

## A  Organization of the Appendix

We organize the appendix as follows:

- The proof details of Theorem 2.3 and discussion of Remark 2.4 are given in Section B.
- The proof details of the algorithm, including the properties of $p$, the convergence analysis of Newton's method, and Bregman proximal alternating linearized minimization method are collected in C.
- Additional experiment results are summarized in Section D.

## B  Proof of Robust Guarantee in Section 2.2

### B.1  Proof of Theorem 2.3

*Proof.* $\text{GW}^{\text{rob}}_{\rho_1,\rho_2}(\mu,\nu)$ is defined as

$$\text{GW}^{\text{rob}}_{\rho_1,\rho_2}(\mu,\nu) = \inf_{\alpha\in\mathcal{P}(X),\ \beta\in\mathcal{P}(Y)}\ \inf_{\pi\in\mathcal{M}^+(X\times Y)} \iint |d_X(x,x') - d_Y(y,y'))|^2 d\pi(x,y)d\pi(x',y') +$$
$$\tau_1 d_{\textbf{KL}}(\pi_1,\alpha) + \tau_2 d_{\textbf{KL}}(\pi_2,\beta)$$
$$\text{s.t.}\quad d_{\textbf{KL}}(\mu,\alpha) \le \rho.$$

Consider $\pi_c$, the optimal transport plans for $\text{GW}(\mu_c,\nu_c)$. Notably, $\pi_c$ serves as a feasible solution for $\text{GW}^{\text{rob}}_{\rho_1,\rho_2}(\mu,\nu)$. Consequently, we can deduce that:

$$\text{GW}^{\text{rob}}_{\rho_1,\rho_2}(\mu,\nu) \le \inf_{\alpha\in\mathcal{P}(X),\ \beta\in\mathcal{P}(Y)} \iint |d_X(x,x') - d_Y(y,y'))|^2 d\pi_c(x,y)d\pi_c(x',y') +$$
$$\tau_1 d_{\textbf{KL}}((\pi_c)_1,\alpha) + \tau_2 d_{\textbf{KL}}((\pi_c)_2,\beta)$$
$$\text{s.t. } d_{\textbf{KL}}(\mu,\alpha) \le \rho_1,\ d_{\textbf{KL}}(\nu,\beta) \le \rho_2$$
$$= \text{GW}(\mu_c,\nu_c) + \inf_{\alpha\in\mathcal{P}(X),\ d_{\textbf{KL}}(\mu,\alpha)\le\rho_1} d_{\textbf{KL}}(\mu_c,\alpha) + \inf_{\beta\in\mathcal{P}(Y),\ d_{\textbf{KL}}(\nu,\beta)\le\rho_2} d_{\textbf{KL}}(\nu_c,\beta).$$

To establish an upper bound for $\text{GW}^{\text{rob}}_{\rho_1,\rho_2}(\mu,\nu)$, let us begin by addressing the following problem:

$$\inf_{\alpha\in\mathcal{P}(X),\ d_{\textbf{KL}}(\mu,\alpha)\le\rho_1} d_{\textbf{KL}}(\mu_c,\alpha) \tag{8}$$

We consider the distribution of the form $(1-\gamma)\mu + \gamma\mu_c$, for $\gamma \in [0,1]$. Then we prove that if $\gamma \le \min\left(\frac{\rho_1}{\epsilon_1 d_{\textbf{KL}}(\mu_a,\mu_c)}, 1\right)$, then $(1-\gamma)\mu + \gamma\mu_c$ is a feasible solution for problem (8).

By the joint convexity of KL divergence, we have

$$d_{\textbf{KL}}(\mu, (1-\gamma)\mu + \gamma\mu_c) \le \gamma d_{\textbf{KL}}(\mu,\mu_c)$$
$$= \gamma d_{\textbf{KL}}((1-\epsilon_1)\mu_c + \epsilon_1\mu_a, \mu_c)$$
$$\le \gamma\epsilon_1 d_{\textbf{KL}}(\mu_a,\mu_c)$$
$$\le \rho_1.$$

Therefore,

$$d_{\textbf{KL}}(\mu_c, (1-\gamma)\mu + \gamma\mu_c) \le (1-\gamma)d_{\textbf{KL}}(\mu_c,\mu)$$
$$= (1-\gamma)d_{\textbf{KL}}(\mu_c, (1-\epsilon_1)\mu_c + \epsilon_1\mu_a)$$
$$\le (1-\gamma)\epsilon_1 d_{\textbf{KL}}(\mu_c,\mu_a)$$

The largest value $\gamma$ can take is $\frac{\rho_1}{\epsilon_1 d_{\textbf{KL}}(\mu_a,\mu_c)}$. This gives

$$\inf_{\alpha\in\mathcal{P}(X),\ d_{\textbf{KL}}(\mu,\alpha)\le\rho_1} d_{\textbf{KL}}(\mu_c,\alpha) \le \max\left(0, 1 - \frac{\rho_1}{\epsilon_1 d_{\textbf{KL}}(\mu_a,\mu_c)}\right)\epsilon_1 d_{\textbf{KL}}(\mu_c,\mu_a).$$

Similarly, we can prove that

$$\inf_{\beta\in\mathcal{P}(Y),\ d_{\textbf{KL}}(\nu,\beta)\le\rho_2} d_{\textbf{KL}}(\nu_c,\beta) \le \max\left(0, 1 - \frac{\rho_2}{\epsilon_2 d_{\textbf{KL}}(\nu_a,\nu_c)}\right)\epsilon_2 d_{\textbf{KL}}(\nu_c,\nu_a).$$

This completes the proof. $\qquad\square$

## B.2 Proof and Discussion of Remark 2.4

Utilizing the same notations as in the proof of Theorem 2.3, and considering that $\pi_c$ is a feasible solution to the UGW problem, substituting it directly leads to the desired result.

Furthermore, referring to [40, Theorem 1] and adapting the notations to our framework, we have the following property: specifically, when considering only $\mu$ is corrupted with outliers, characterized by $\mu = (1 - \epsilon_1)\mu_c + \epsilon_1\mu_a$, while $\nu = \nu_c$. If we let $\delta = 2(\tau_1 + \tau_2)\epsilon_1$, and $K = M + \frac{1}{M}\text{UGW}(\mu_c, \nu) + \delta$, where $M$ represents the transported mass between clean data and $\Delta_\infty$ signifies the maximal deviation between the contaminated source and the target, then the following inequality holds:

$$\text{UGW}(\mu, \nu) \leq (1 - \epsilon_1)\text{UGW}(\mu_c, \nu) + \delta M \left[1 - \exp\left(-\frac{\Delta_\infty(1 + M) + K}{\delta M}\right)\right].$$

Given that $\text{UGW}(\mu_c, \nu) \leq \text{GW}(\mu_c, \nu)$, we can derive the following relationship:

$$\text{UGW}(\mu, \nu) \leq (1 - \epsilon_1)\text{GW}(\mu_c, \nu) + \delta M \left[1 - \exp\left(-\frac{\Delta_\infty(1 + M) + M + \frac{1}{M}\text{GW}(\mu_c, \nu) + \delta}{\delta M}\right)\right]. \quad (9)$$

Comparing this result to Remark 2.4, (9) also incorporates $\text{GW}(\mu_c, \nu)$ in the exponential term and $M$, the transported mass between clean data. These variables may be influenced by the marginal parameters $\tau_1$ and $\tau_2$. As a result, finding an optimal choice for $\tau_1$ and $\tau_2$ to establish a tight upper bound for UGW using the GW distance between clean samples, as presented in (9), proves challenging. As outlined in Remark 2.4, it is worth noting that setting $\tau_1 = \tau_2 = 0$ represents the sole means to attain a tight upper bound. Nevertheless, in real-world applications, this approach is impractical, as it could lead to over-relaxation and compromise the performance.

# C  Proof Details of Bregman Proximal Alternating Linearized Minimization Method for Robust GW

Given a vector x, we use $\|x\|_2$ to denote its $\ell_2$ norm. We use $\|X\|_F$ to denote the Frobenius norm of matrix $X$. For a convex set $C$ and a point $x$, we define the distance between $C$ and $x$ as

$$\text{dist}(x, C) = \min_{y \in C} \|x - y\|_2.$$

## C.1  Proof of Proposition 3.1

*Proof.* Problem (7) can be written as

$$\min_{\alpha \in \Delta^n} \sum_{i=1}^{n} \left( (\pi_1^{k+1})_i \log\left(\frac{(\pi_1^{k+1})_i}{\alpha_i}\right) - (\pi_1^{k+1})_i + \alpha_i \right) + \frac{1}{c_k} \sum_{i=1}^{k} \alpha_i^k \log\left(\frac{\alpha_i^k}{\alpha_i}\right) + w \left( \sum_{i=1}^{n} \mu_i \log\left(\frac{\mu_i}{\alpha_i}\right) - \rho_1 \right).$$

We first consider a relaxed problem of the problem above:

$$\min_{\alpha^T \mathbf{1}_n = 1} \sum_{i=1}^{n} \left( (\pi_1^{k+1})_i \log\left(\frac{(\pi_1^{k+1})_i}{\alpha_i}\right) - (\pi_1^{k+1})_i + \alpha_i \right) + \frac{1}{c_k} \sum_{i=1}^{n} \alpha_i^k \log\left(\frac{\alpha_i^k}{\alpha_i}\right) + w \left( \sum_{i=1}^{n} \mu_i \log\left(\frac{\mu_i}{\alpha_i}\right) - \rho_1 \right).$$

Consider the Lagrangian function of the relaxed problem

$$L(\alpha, \lambda) = \sum_{i=1}^{n} \left( (\pi_1^{k+1})_i \log\left(\frac{(\pi_1^{k+1})_i}{\alpha_i}\right) - (\pi_1^{k+1})_i + \alpha_i \right) + \frac{1}{c_k} \sum_{i=1}^{n} \alpha_i^k \log\left(\frac{\alpha_i^k}{\alpha_i}\right) +$$

$$w \left( \sum_{i=1}^{n} \mu_i \log\left(\frac{\mu_i}{\alpha_i}\right) - \rho_1 \right) + \lambda \left(\alpha^T \mathbf{1}_n - 1\right).$$

Let

$$\frac{\partial L}{\partial \alpha} = \begin{pmatrix} -\frac{(\pi_1^{k+1})_1}{\alpha_1} + 1 \\ -\frac{(\pi_1^{k+1})_2}{\alpha_2} + 1 \\ \vdots \\ -\frac{(\pi_1^{k+1})_n}{\alpha_n} + 1 \end{pmatrix} + \frac{1}{c_k} \begin{pmatrix} -\frac{\alpha_1^k}{\alpha_1} \\ -\frac{\alpha_2^k}{\alpha_2} \\ \vdots \\ -\frac{\alpha_n^k}{\alpha_n} \end{pmatrix} + w \begin{pmatrix} -\frac{\mu_1}{\alpha_1} \\ -\frac{\mu_2}{\alpha_2} \\ \vdots \\ -\frac{\mu_n}{\alpha_n} \end{pmatrix} + \begin{pmatrix} \lambda \\ \lambda \\ \vdots \\ \lambda \end{pmatrix} = 0.$$

Then we obtain that
$$\alpha_i = \frac{(\pi_1^{k+1})_i + \frac{1}{c_k}\alpha_i^k + w\mu_i}{1+\lambda}.$$

Since $\sum_{i=1}^n \alpha_i = 1$, $\sum_{i=1}^n \alpha_i^k = 1$, and $\sum_{i=1}^n \mu_i = 1$, then $1+\lambda = \sum_{ij}\pi_{ij}^{k+1} + \frac{1}{c_k} + w$. Thus, the optimal solution to the relaxed problem is
$$\hat{\alpha}(w) = \frac{\pi^{k+1}\mathbf{1}_m + \frac{1}{c_k}\alpha^k + w\mu}{\sum_{i,j}\pi_{ij}^{k+1} + \frac{1}{c_k} + w}.$$

We can see that $\hat{\alpha}(w) \geq 0$. Hence, $\hat{\alpha}(w)$ is also the optimal solution to problem (7).

Since if $w$ satisfies (i) or (ii), $(\hat{\alpha}(w), w)$ is a solution to KKT conditions of problem (5), therefore, $\hat{\alpha}(w)$ is an optimal solution to problem (5). Next, we prove that $p$ is differentiable when $h$ is relative entropy. Problem (7) admits the closed-form solution
$$\hat{\alpha}(w) = \frac{\pi^{k+1}\mathbf{1}_m + \frac{1}{c_k}\alpha^k + w\mu}{\sum_{i,j}\pi_{ij}^{k+1} + \frac{1}{c_k} + w}. \tag{10}$$

By substituting (10) into $p$, $p(w)$ can be written as
$$p(w) = \sum_{i=1}^n \mu_i \log\left(\frac{\mu_i\left(\sum_{i,j}\pi_{ij}^{k+1} + \frac{1}{c_k} + w\right)}{(\pi^{k+1}\mathbf{1}_m)_i + \frac{1}{c_k}\alpha_i^k + w\mu_i}\right) - \rho_1.$$

Thus, $p$ is twice differentiable. The first-order derivative and second-order of $p$ are
$$p'(w) = \sum_{i=1}^n \mu_i \frac{\left((\pi^{k+1}\mathbf{1}_m)_i + \frac{1}{c_k}\alpha_i^k + \mu_i w\right) - \mu_i\left(\sum_{ij}\pi_{ij}^{k+1} + \frac{1}{c_k} + w\right)}{\left(\sum_{ij}\pi_{ij}^{k+1} + \frac{1}{c_k} + w\right)\left((\pi^{k+1}\mathbf{1}_m)_i + \frac{1}{c_k}\alpha_i^k + \mu_i w\right)},$$

and
$$p''(w) = -\sum_{i=1}^n \mu_i \frac{\left((\pi^{k+1}\mathbf{1}_m)_i + \frac{1}{c_k}\alpha_i^k + \mu_i w\right)^2 - \mu_i^2\left(\sum_{ij}\pi_{ij}^{k+1} + \frac{1}{c_k} + w\right)^2}{\left(\sum_{ij}\pi_{ij}^{k+1} + \frac{1}{c_k} + w\right)^2\left((\pi^{k+1}\mathbf{1}_m)_i + \frac{1}{c_k}\alpha_i^k + \mu_i w\right)^2}.$$

Then we prove that $p'(w) \leq 0$ and $p''(w) \geq 0$ for $w \geq 0$, so $p$ is monotonically non-increasing and convex on $\mathbb{R}_+$. Let $s_i = (\pi^{k+1}\mathbf{1}_m)_i + \frac{1}{c_k}\alpha_i^k + \mu_i w$ and $s = (\pi^{k+1}\mathbf{1}_m)_i + \frac{1}{c_k}\alpha_i^k + \mu_i w$. Note that $s = \sum_{i=1}^n s_i$. Then $p'$ and $p''$ can be written as
$$p'(w) = \sum_{i=1}^n \mu_i \frac{s_i - \mu_i s}{s_i \cdot s} = \frac{1}{s}\left(1 - \sum_{i=1}^n \mu_i \frac{\mu_i s}{s_i}\right),$$

and
$$p''(w) = -\sum_{i=1}^n \mu_i \frac{s_i^2 - \mu_i^2 s^2}{s_i^2 \cdot s^2} = -\frac{1}{s^2}\left(1 - \sum_{i=1}^n \mu_i \frac{\mu_i^2 s^2}{s_i^2}\right).$$

Therefore, it is equivalent to show $\sum_{i=1}^n \mu_i \frac{\mu_i s}{s_i} \geq 1$ and $\sum_{i=1}^n \mu_i \frac{\mu_i^2 s^2}{s_i^2} \geq 1$.

Recall that $\frac{1}{x}$ and $\frac{1}{x^2}$ are convex on $\mathbb{R}_{++}$, then
$$\sum_{i=1}^n \mu_i \frac{\mu_i s}{s_i} = \sum_{i=1}^n \frac{1}{\mu_i \frac{s_i}{\mu_i s}} \geq \frac{1}{\sum_{i=1}^n \mu_i \frac{s_i}{\mu_i s}} = 1,$$

$$\sum_{i=1}^n \mu_i \frac{\mu_i^2 s^2}{s_i^2} = \sum_{i=1}^n \mu_i \frac{1}{\left(\frac{s_i}{\mu_i s}\right)^2} \geq \frac{1}{\left(\sum_{i=1}^n \mu_i \frac{s_i}{\mu_i s}\right)^2} = 1.$$

$\square$

## C.2 Proof of Proposition 3.2

*Proof.* First, prove that $p$ only has one root on $I$. Since $p$ is continuous on $I$ and there exists $\tilde{x}, \bar{x} \in I$ such that $p(\tilde{x}) > 0$ and $p(\bar{x}) < 0$, $p$ contains at least one root on $[\tilde{x}, \bar{x}]$. Since $p$ is non-increasing, $p$ cannot have roots outside $[\tilde{x}, \bar{x}]$. Suppose that $p$ have two different roots $z_1$ and $z_2$ on $[\tilde{x}, \bar{x}]$ and $z_1 < z_2$. By convexity of $p$, we have

$$0 = p(z_2) = p\left(\frac{\bar{x} - z_2}{\bar{x} - z_1}z_1 + \frac{z_2 - z_1}{\bar{x} - z_1}\bar{x}\right) \leq \frac{\bar{x} - z_2}{\bar{x} - z_1}p(z_1) + \frac{z_2 - z_1}{\bar{x} - z_1}p(\bar{x}) = \frac{z_2 - z_1}{\bar{x} - z_1}p(\bar{x}) < 0.$$

This is a contradiction. So $p$ has a unique root on $I$.

$p'(x) \leq 0$ since $p$ is non-increasing on $I$. Denote the root of $p$ as $r$. Claim that $p'(x) < 0$ for $x \in [\tilde{x}, r]$. Otherwise, there exist $x \in [\tilde{x}, r]$ such that $p'(x) = 0$, then

$$0 > p(\bar{x}) \geq p(x) + p'(x)(\bar{x} - x) = p(x) \geq 0,$$

which leads to a contradiction. Especially, $p'(\tilde{x}) < 0$, and we can set $\tilde{x}$ as the initial point of Newton's method.

The update of Newton's method is

$$x_{k+1} = x_k - \frac{p(x_k)}{p'(x_k)}.$$

Therefore, $x_{k+1} \geq x_k$ and $\{x_k\}_{k \geq 1}$ is an increasing sequence. Since $p$ is convex,

$$p(x_{k+1}) \geq p(x_k) + p'(x_k)(x_{k+1} - x_k) = p(x_k) - p(x_k) = 0.$$

$x_k \leq r$ because $p$ is a monotonically non-increasing function. $\{x_k\}_{k \geq 1}$ is an increasing sequence with an upper bound, so it has a limit $x^*$ and $\lim_{k\to\infty}(x_k - x_{k+1}) = 0$. Also, $p'$ is bounded on $[\tilde{x}, r]$ since it is continuous. Therefore,

$$p(x^*) = \lim_{k \to +\infty} p(x_k) = \lim_{k \to +\infty} p'(x_k)(x_k - x_{k+1}) = 0.$$

Hence, the sequence generated by Newton's method converges to a root of $p$. $\qquad\square$

## C.3 Proof of Theorem 3.3

**Assumption C.1.** The critical point set $\mathcal{X}$ is non-empty.

Before the proof of Theorem 3.3, we first prove that sequence $\{\pi^k\}_{k \geq 0}$ generated by BPALM lies in a compact set.

**Proposition C.2.** *Sequence $\{\pi^k\}_{k \geq 0}$ generated by BPALM lies in a compact set.*

*Proof.* We prove that $\{\pi^k\}_{k \geq 0}$ lies in the compact set $\mathcal{A} := \{\pi \in \mathbb{R}^{n \times m} : 0 \leq \pi_{ij} \leq 1\}$ by mathematical induction. For $k = 0$, we can initialize $\pi^0$ with $0 \leq \pi_{ij}^0 \leq 1$. Suppose that $\pi^k \in \mathcal{A}$ and $\pi^{k+1} \notin \mathcal{A}$. Then there exist $i \in [n]$ and $j \in [m]$ such that $\pi_{ij}^{k+1} > 1$. Recall that $\pi^{k+1}$ is the optimal solution to the problem

$$\min_{\pi \geq 0} \varphi(\pi) := \langle \mathcal{L}(D, \bar{D}) \otimes \pi^k, \pi \rangle + \tau_1 d_{\mathbf{KL}}(\pi \mathbf{1}_m, \alpha^k) + \tau_2 d_{\mathbf{KL}}(\pi^T \mathbf{1}_n, \beta^k) + \frac{1}{t_k}d_{\mathbf{KL}}(\pi, \pi^k), \quad (11)$$

Observe that function $\phi(x) := x \log \frac{x}{a} - x + a$ is a unimodal function on $\mathbb{R}_+$ and achieve its minimum at $x = a$. Since $\alpha_i$, $\beta_j$, and $\pi_{ij}^k$ are smaller than or equal to 1, $\alpha_i$, $\beta_j$, and $\pi_{ij}^k$ are strictly smaller than $\pi_{ij}^{k+1}$. Let $\tilde{\pi} \in \mathbb{R}^{n \times m}$,

$$\tilde{\pi}_{kl} = \begin{cases} \max\{\alpha_i, \beta_j, \pi_{ij}^k\}, & (k, l) = (i, j), \\ \pi_{kl}^{k+1}, & \text{otherwise.} \end{cases}$$

Then $\varphi(\tilde{\pi}) < \varphi(\pi^k)$, this contradicts to $\pi^{k+1}$ is the optimal solution to problem (11). Thus, $\pi^{k+1} \in \mathcal{A}$. $\qquad\square$

For the proof of Theorem 3.3, we first prove the sufficient decrease property of BPALM, i.e., there exist a constant $\kappa_1 > 0$ and an index $k_1 \geq 0$ such that for $k \geq k_1$,

$$F\left(\pi^{k+1}, \alpha^{k+1}, \beta^{k+1}\right) - F\left(\pi^k, \alpha^k, \beta^k\right) \leq -\kappa_1 \left(\left\|\pi^{k+1} - \pi^k\right\|_F^2 + \left\|\alpha^{k+1} - \alpha^k\right\|_2^2 + \left\|\beta^{k+1} - \beta^k\right\|_2^2\right).$$

And then we prove the subsequence convergence result.

*Proof.* It is worth noting that $f(\pi)$ is a quadratic function, i.e., $f(\pi) = \left\langle \mathcal{L}(D, \bar{D}) \otimes \pi, \pi \right\rangle$, then $f(\pi)$ is gradient Lipschitz continuous with the constant $\max_{i,j} \left(\sum_{k,l} \mathcal{L}(D, \bar{D})_{i,j,k,l}^2\right)^{1/2}$. To simplify the notation, let $L_f = \max_{i,j} \left(\sum_{k,l} \mathcal{L}(D, \bar{D})_{i,j,k,l}^2\right)^{1/2}$.

$$F\left(\pi^{k+1}, \alpha^{k+1}, \beta^{k+1}\right) - F\left(\pi^k, \alpha^k, \beta^k\right)$$

$$\leq \left\langle \nabla f(\pi^k), \pi^{k+1} - \pi^k \right\rangle + \frac{L_f}{2}\left\|\pi^{k+1} - \pi^k\right\|_F^2 + q\left(\pi^{k+1}\right) + g_1\left(\pi^{k+1}, \alpha^{k+1}\right) + g_2\left(\pi^{k+1}, \beta^{k+1}\right) +$$
$$h_1\left(\alpha^{k+1}\right) + h_2\left(\beta^{k+1}\right) - \left(q\left(\pi^k\right) + g_1\left(\pi^k, \alpha^k\right) + g_2\left(\pi^k, \beta^k\right) + h_1\left(\alpha^k\right) + h_2\left(\beta^k\right)\right)$$

$$\overset{(\Diamond)}{\leq} \left\langle \nabla f(\pi^k), \pi^{k+1} - \pi^k \right\rangle + \frac{L_f}{\sigma} d_{\mathbf{KL}}\left(\pi^{k+1}, \pi^k\right) + q\left(\pi^{k+1}\right) + g_1(\pi^{k+1}, \alpha^{k+1}) + g_2\left(\pi^{k+1}, \beta^{k+1}\right) +$$
$$h_1\left(\alpha^{k+1}\right) + h_2\left(\beta^{k+1}\right) - \left(q\left(\pi^k\right) + g_1\left(\pi^k, \alpha^k\right) + g_2\left(\pi^k, \beta^k\right) + h_1\left(\alpha^k\right) + h_2\left(\beta^k\right)\right)$$

$$= \left\langle \nabla f(\pi^k), \pi^{k+1} \right\rangle + q\left(\pi^{k+1}\right) + g_1\left(\pi^{k+1}, \alpha^k\right) + g_2\left(\pi^{k+1}, \beta^k\right) + \frac{1}{t_k} d_{\mathbf{KL}}\left(\pi^{k+1}, \pi^k\right) -$$

$$\left\langle \nabla f(\pi^k), \pi^k \right\rangle - q\left(\pi^k\right) - g_1\left(\pi^k, \alpha^k\right) - g_2\left(\pi^k, \beta^k\right) + g_1\left(\pi^{k+1}, \alpha^{k+1}\right) + h_1\left(\alpha^{k+1}\right) + \frac{1}{c_k} d_{\mathbf{KL}}\left(\alpha^k, \alpha^{k+1}\right) -$$

$$g_1\left(\pi^k, \alpha^k\right) - h_1\left(\alpha^k\right) + g_2\left(\pi^{k+1}, \beta^{k+1}\right) + h_2\left(\beta^{k+1}\right) + \frac{1}{r_k} d_{\mathbf{KL}}\left(\beta^k, \beta^{k+1}\right) - g_2\left(\pi^k, \beta^k\right) - h_2\left(\beta^k\right) -$$

$$\left(\frac{1}{t_k} - \frac{L_f}{\sigma}\right) d_{\mathbf{KL}}\left(\pi^{k+1}, \pi^k\right) - \frac{1}{c_k} d_{\mathbf{KL}}\left(\alpha^k, \alpha^{k+1}\right) - \frac{1}{r_k} d_{\mathbf{KL}}\left(\beta^k, \beta^{k+1}\right)$$

$$\leq -\left(\frac{1}{t_k} - \frac{L_f}{\sigma}\right) d_{\mathbf{KL}}\left(\pi^{k+1}, \pi^k\right) - \frac{1}{c_k} d_{\mathbf{KL}}\left(\alpha^k, \alpha^{k+1}\right) - \frac{1}{r_k} d_{\mathbf{KL}}\left(\beta^k, \beta^{k+1}\right).$$

$$\overset{(\Diamond)}{\leq} -\frac{\sigma}{2}\left(\left(\frac{1}{t_k} - \frac{L_f}{\sigma}\right) \left\|\pi^{k+1} - \pi^k\right\|_F^2 + \frac{1}{c_k}\left\|\alpha^{k+1} - \alpha^k\right\|_2^2 + \frac{1}{r_k}\left\|\beta^{k+1} - \beta^k\right\|_2^2\right).$$

$(\Diamond)$ is because as $x \log x$ is $\sigma$-strongly convex, we have

$$d_{\mathbf{KL}}(\pi^{k+1}, \pi^k) \geq \frac{\sigma}{2}\|\pi^{k+1} - \pi^k\|_F^2, \quad d_{\mathbf{KL}}(\alpha^k, \alpha^{k+1}) \geq \frac{\sigma}{2}\|\alpha^{k+1} - \alpha^k\|_2^2, \quad d_{\mathbf{KL}}(\beta^k, \beta^{k+1}) \geq \frac{\sigma}{2}\|\beta^{k+1} - \beta^k\|_2^2.$$

By letting $\kappa_1 = \frac{\sigma}{2} \min\left(\left(\frac{1}{t_k} - \frac{L_f}{\sigma}\right), \frac{1}{\bar{r}}\right) > 0$, we get

$$F\left(\pi^{k+1}, \alpha^{k+1}, \beta^{k+1}\right) - F\left(\pi^k, \alpha^k, \beta^k\right) \leq -\kappa_1 \left(\left\|\pi^{k+1} - \pi^k\right\|_F^2 + \left\|\alpha^{k+1} - \alpha^k\right\|_2^2 + \left\|\beta^{k+1} - \beta^k\right\|_2^2\right).$$
$$(12)$$

Summing up (12) from $k = 0$ to $+\infty$, we obtain

$$\kappa_1 \sum_{k=0}^{\infty} \left(\left\|\pi^{k+1} - \pi^k\right\|_F^2 + \left\|\alpha^{k+1} - \alpha^k\right\|_2^2 + \left\|\beta^{k+1} - \beta^k\right\|_2^2\right) \leq F(\pi^0, \alpha^0, \beta^0) - F(\pi^\infty, \alpha^\infty, \beta^\infty).$$

As $F$ is coercive and $\left\{\left(\pi^k, \alpha^k, \beta^k\right)\right\}$ is a bounded sequence, it follows that the left-hand side is bounded. This implies

$$\sum_{k=0}^{\infty} \left(\left\|\pi^{k+1} - \pi^k\right\|_F^2 + \left\|\alpha^{k+1} - \alpha^k\right\|_2^2 + \left\|\beta^{k+1} - \beta^k\right\|_2^2\right) < +\infty,$$

and

$$\lim_{k \to +\infty} \left(\|\pi^{k+1} - \pi^k\|_F + \|\alpha^{k+1} - \alpha^k\|_2 + \|\beta^{k+1} - \beta^k\|_2\right) = 0.$$

Let $l(x) = \sum_i x_i \log x_i$. Recall the optimality condition of BPALM, we have

$$0 \in \nabla f\left(\pi^{k+1}\right) + \partial q\left(\pi^{k+1}\right) + \nabla_\pi g_1\left(\pi^{k+1}, \alpha^k\right) + \nabla_\pi g_2\left(\pi^{k+1}, \beta^k\right) + \frac{1}{t_k}\left(\nabla l\left(\pi^{k+1}\right) - \nabla l\left(\pi^k\right)\right),$$
(13)

$$0 \in \nabla_\alpha g_1\left(\pi^{k+1}, \alpha^{k+1}\right) + \partial h_1\left(\alpha^{k+1}\right) + \frac{1}{c_k}\nabla^2 l(\alpha^{k+1})(\alpha^{k+1} - \alpha^k),$$
(14)

$$0 \in \nabla_\beta g_2\left(\pi^{k+1}, \beta^{k+1}\right) + \partial h_2\left(\beta^{k+1}\right) + \frac{1}{r_k}\nabla^2 l(\beta^{k+1})(\beta^{k+1} - \beta^k).$$
(15)

Let $(\pi^\infty, \alpha^\infty, \beta^\infty)$ be a limit point of the sequence $\left\{(\pi^k, \alpha^k, \beta^k)\right\}_{k\geq 0}$. Then, there exists a sequence $\{n_k\}_{k\geq 0}$ such that $\left\{(\pi^{n_k}, \alpha^{n_k}, \beta^{n_k})\right\}_{k\geq 0}$ converges to $(\pi^\infty, \alpha^\infty, \beta^\infty)$. Since we assume that $h$ is twice continuous differentiable and $\alpha^k$ and $\beta^k$ are in a compact set, then $\nabla^2 l(\alpha^k)$ and $\nabla^2 l(\beta^k)$ are bounded. Therefore, $\lim_{k\to\infty} \nabla^2 l(\alpha^{k+1})(\alpha^{k+1} - \alpha^k) = 0$ and $\lim_{k\to\infty} \nabla^2 l(\beta^{k+1})(\beta^{k+1} - \beta^k) = 0$. Replacing the $k$ by $n_k$ in (13), (14), and (15), taking limits on both sides as $k \to \infty$, we obtain that

$$0 \in \nabla f\left(\pi^\infty\right) + \partial q(\pi^\infty) + \nabla_\pi g_1\left(\pi^\infty, \alpha^\infty\right) + \nabla_\pi g_2\left(\pi^\infty, \beta^\infty\right),$$
$$0 \in \nabla_\alpha g_1\left(\pi^\infty, \alpha^\infty\right) + \partial h_1\left(\alpha^\infty\right),$$
$$0 \in \nabla_\beta g_2\left(\pi^\infty, \beta^\infty\right) + \partial h_2\left(\beta^\infty\right).$$

Thus $(\pi^\infty, \alpha^\infty, \beta^\infty)$ belongs to $\mathcal{X}$. $\qquad\square$

### C.4 Discussion of Computational Complexity of PGW, UGW, and RGW

We consider the measure $\min_{0\leq k\leq K}(\|\pi^{k+1} - \pi^k\|_F + \|\alpha^{k+1} - \alpha^k\|_2 + \|\beta^{k+1} - \beta^k\|_2)$ as the stationary measure, and it is observed that the convergence rate of our algorithm is $\mathcal{O}(\frac{1}{\sqrt{K}})$. Similarly, the Frank-Wolfe algorithm for PGW also exhibits a convergence rate of $\mathcal{O}(\frac{1}{\sqrt{K}})$. The literature on UGW does not provide a discussion on the convergence rate of alternate Sinkhorn minimization for UGW. In each iteration of PGW, UGW, and RGW, the computation of $\mathcal{L}(D, \bar{D}) \otimes \pi^k$ is a crucial step. According to [31], the complexity of this computation is $\mathcal{O}(n^2 m + m^2 n)$. Additionally, PGW involves utilizing the network simplex algorithm to solve a linear programming problem as a subroutine, which has a complexity of $\mathcal{O}((n^2 m + m^2 n)\log^2(n + m))$. On the other hand, both UGW and RGW utilize the sinkhorn algorithm to solve an entropic unbalanced optimal transport problem. The complexity of the sinkhorn algorithm for unbalanced OT is $\mathcal{O}((n^2 + m^2)/(\varepsilon \log(\varepsilon)))$ for computing an $\varepsilon$-approximation.

## D  Additional Experiment Results

### D.1  Additional Experiment Results on Subgraph Alignment

**Source codes of all baselines used in this paper:**

- FW [17]: https://github.com/PythonOT/POT
- SpecGW [15]: https://github.com/trneedham/Spectral-Gromov-Wasserstein
- eBPG [17]: https://github.com/PythonOT/POT
- BAPG [24]: https://github.com/squareRoot3/Gromov-Wasserstein-for-Graph
- UGW [35]: https://github.com/thibsej/unbalanced_gromov_wasserstein
- PGW [10, 17]: https://github.com/PythonOT/POT
- srGW: [42]: https://github.com/cedricvincentcuaz/srGW
- RGWD: [25]: https://github.com/cxxszz/rgdl

**Results in Figure 4**  The data utilized to create Figure 4 is provided in Table 2 and Table 3.

Table 2: Comparison of the average matching accuracy (%) and wall-clock time (seconds) on subgraph alignment of 30% subgraph and 20% subgraph.

| | 30% subgraph | | | | | | 20% subgraph | | | | | |
| | Synthetic | | Proteins | | Enzymes | | Synthetic | | Proteins | | Enzymes | |
| Method | Acc | Time | Acc | Time | Acc | Time | Acc | Time | Acc | Time | Acc | Time |
|---|---|---|---|---|---|---|---|---|---|---|---|---|
| FW | 2.22 | 17.06 | 12.96 | 14.83 | 12.08 | 5.37 | 2.24 | 6.65 | 10.83 | 11.34 | 9.53 | 4.92 |
| SpecGW | 1.38 | 2.24 | 10.64 | 11.54 | 9.41 | 3.74 | 1.71 | 2.21 | 10.78 | 10.15 | 8.35 | 3.21 |
| eBPG | 0.65 | 0.49 | 8.12 | 1022.02 | 3.84 | 476.83 | 1.17 | 0.42 | 7.23 | 545.50 | 2.66 | 94.78 |
| BPG | 1.86 | 17.53 | 17.89 | 86.85 | 17.69 | 52.89 | 1.64 | 11.66 | 12.99 | 55.47 | 14.35 | 32.89 |
| BAPG | 2.94 | 35.90 | 18.79 | 36.02 | 16.85 | 10.88 | 3.80 | 23.29 | 14.07 | 23.92 | 11.22 | 8.38 |
| srGW | 3.17 | 86.38 | 22.75 | 89.14 | 27.45 | 41.18 | 5.49 | 88.89 | 18.38 | 17.72 | 23.13 | 17.11 |
| RGWD | 1.94 | 933.25 | 16.90 | 3674.98 | 16.34 | 3322.16 | 1.94 | 933.25 | 16.90 | 3674.98 | 16.34 | 3322.16 |
| UGW | 35.15 | 168.03 | 14.32 | 10298 | 10.91 | 5552.27 | 4.48 | 251.41 | 11.75 | 7813.96 | 10.40 | 4019.62 |
| PGW | 2.06 | 339.23 | 11.68 | 507.11 | 11.77 | 174.26 | 1.90 | 227.87 | 9.34 | 365.88 | 7.97 | 165.27 |
| RGW | **52.35** | 679.00 | **30.17** | 947.48 | **37.12** | 538.04 | **11.58** | 229.05 | **23.51** | 546.15 | **25.39** | 879.93 |

Table 3: Comparison of the average matching accuracy (%) and wall-clock time (seconds) on subgraph alignment of 40% subgraph.

| | 40% subgraph | | | | | |
| | Synthetic | | Proteins | | Enzymes | |
| Method | Acc | Time | Acc | Time | Acc | Time |
|---|---|---|---|---|---|---|
| FW | 1.84 | 17.96 | 15.34 | 19.64 | 14.22 | 6.36 |
| SpecGW | 1.72 | 3.25 | 11.21 | 12.17 | 9.59 | 3.88 |
| eBPG | 0.38 | 0.51 | 12.16 | 1628.38 | 9.96 | 943.49 |
| BPG | 3.41 | 18.61 | 24.31 | 108.10 | 24.58 | 62.81 |
| BAPG | 7.61 | 22.55 | 23.78 | 36.81 | 24.82 | 11.13 |
| srGW | 2.45 | 120.12 | 22.58 | 74.58 | 27.02 | 32.14 |
| RGWD | 3.48 | 930.84 | 22.73 | 4490.60 | 22.63 | 3205.03 |
| UGW | 79.61 | 960.04 | 21.22 | 11398 | 22.26 | 5589.73 |
| PGW | 2.17 | 483.10 | 11.95 | 491.64 | 9.51 | 182.58 |
| RGW | **90.79** | 662.15 | **38.94** | 769.25 | **48.11** | 291.74 |

**Selection of Stepsize $t_k$, $c_k$, and $r_k$** In the subgraph alignment task, we have used constant values for the stepsizes $t_k$, $c_k$ and $r_k$. We have conducted a sensitivity analysis for these parameters, and the details are summarized in Tables 4 and 5. Specifically, Table 4 reveals that RGW achieves its highest accuracy with $t$ in the range of 0.01 to 0.05, allowing us to select $t = 0.01$ as the default. Table 5 further indicates that accuracy is not significantly affected by variations in $c$, leading us to set $c = 0.1$ as the default.

Table 4: The performance of RGW with different stepsize $t$ on three subgraph alignment databases.

| | Synthetic | | Proteins-1 | | Enzymes | |
| RGW | Acc | Time | Acc | Time | Acc | Time |
|---|---|---|---|---|---|---|
| $t = 0.01$ | 94.64 | 541.50 | 53.07 | 567.09 | 63.23 | 139.82 |
| $t = 0.05$ | 90.49 | 288.25 | 53.37 | 1030.15 | 63.69 | 304.11 |
| $t = 0.1$ | 87.00 | 233.65 | 53.69 | 713.07 | 62.38 | 506.83 |
| $t = 0.5$ | 87.09 | 779.84 | 51.56 | 1797.13 | 60.07 | 822.23 |
| $t = 1.0$ | 71.10 | 491.93 | 50.21 | 3071.27 | 58.31 | 1166.88 |

**Experiment Results of Normalized Degree as Marginal Distribution** In addition to employing the uniform distribution as node distribution, we also explore the use of normalized degrees as node distribution. The results presented in Table 6 confirm that RGW surpasses other methods in terms of accuracy when utilizing normalized degrees as marginal distributions.

**Experiment Results of Adding Noise to Query Graph** We conducted an experiment by adding 10% pseudo edges to the subgraphs, and the results can be found in Table 7. These findings

Table 5: The performance of RGW with different stepsize $c$ on three subgraph alignment databases.

| RGW | Synthetic | | Proteins-1 | | Enzymes | |
|---|---|---|---|---|---|---|
| | Acc | Time | Acc | Time | Acc | Time |
| $c = 0.01$ | 90.39 | 481.06 | 53.17 | 459.43 | 63.06 | 163.45 |
| $c = 0.05$ | 90.39 | 967.98 | 53.12 | 919.18 | 62.86 | 327.44 |
| $c = 0.1$ | 90.39 | 1460.94 | 53.37 | 1374.43 | 63.69 | 490.91 |
| $c = 0.5$ | 90.34 | 1969.18 | 53.35 | 1822.70 | 63.36 | 650.25 |
| $c = 1.0$ | 90.28 | 2488.17 | 53.43 | 2262.07 | 63.53 | 807.57 |

Table 6: Subgraph alignment results (Acc.) of 50% subgraph of compared GW-based methods using normalized degree.

| Method | Synthetic | Proteins-1 | Proteins-2 | Enzymes |
|---|---|---|---|---|
| FW | 2.96 | 14.82 | 42.16 | 15.90 |
| SpecGW | 1.57 | 8.92 | 43.10 | 12.07 |
| eBPG | 5.27 | 13.77 | 31.90 | 13.51 |
| BPG | 12.52 | 20.88 | 57.77 | 29.42 |
| BAPG | 74.39 | 24.65 | 63.26 | 31.92 |
| srGW | 4.22 | 13.67 | 12.39 | 23.05 |
| RGWD | 13.54 | 25.92 | 57.30 | 28.12 |
| UGW | 99.56 | 25.51 | 62.92 | 39.71 |
| PGW | 3.97 | 11.59 | 37.79 | 13.08 |
| RGW | **99.61** | **50.61** | **66.09** | **63.59** |

demonstrate that RGW significantly outperforms other methods on the Enzymes and Proteins datasets.

Table 7: Subgraph alignment results (Mean $\pm$ Std.) of 50% subgraph in 5 independent trials over different random seeds in the noise generating process.

| Method | Synthetic | Proteins-1 | Enzymes |
|---|---|---|---|
| FW | 2.38$\pm$0.27 | 10.21$\pm$1.22 | 12.58$\pm$12.58 |
| SpecGW | 1.79$\pm$0.26 | 9.91$\pm$0.56 | 10.00$\pm$0.81 |
| eBPG | 6.84$\pm$1.89 | 15.63$\pm$0.73 | 14.31$\pm$0.77 |
| BPG | 17.71$\pm$2.23 | 20.96$\pm$1.57 | 22.29$\pm$1.19 |
| BAPG | 43.39 $\pm$ 4.84 | 23.08$\pm$1.12 | 24.42$\pm$ 1.54 |
| srGW | 1.75$\pm$ 0.18 | 13.34$\pm$0.56 | 20.03$\pm$ 0.73 |
| RGWD | 17.30$\pm$ 1.90 | 22.70$\pm$0.33 | 24.07$\pm$ 0.44 |
| UGW | 81.24$\pm$ 2.34 | 22.99$\pm$0.16 | 26.68$\pm$ 1.54 |
| PGW | 1.67 $\pm$ 0.43 | 8.19$\pm$0.92 | 7.33$\pm$ 0.45 |
| RGW | **88.79** $\pm$ 1.59 | **38.88**$\pm$1.31 | **49.01**$\pm$ 0.99 |

## D.2    Additional Experiment Results on Partial Shape Correspondence

**Convergence of BPALM**    The convergence results for the proposed BPALM using various step sizes are presented in Figure 7 for the toy example discussed in Section 4.1.

**Additional Experiment Result on TOSCA Dataset**    Additional experiment results on TOSCA Dataset are shown in Figure 8, 9, and 10.

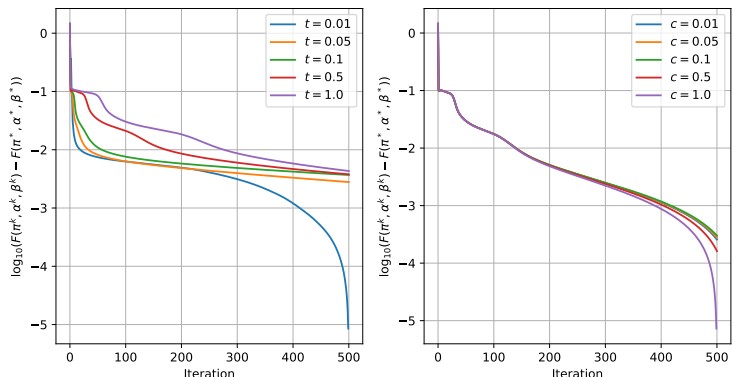

Figure 7: Convergence result of toy example with different stepsizes

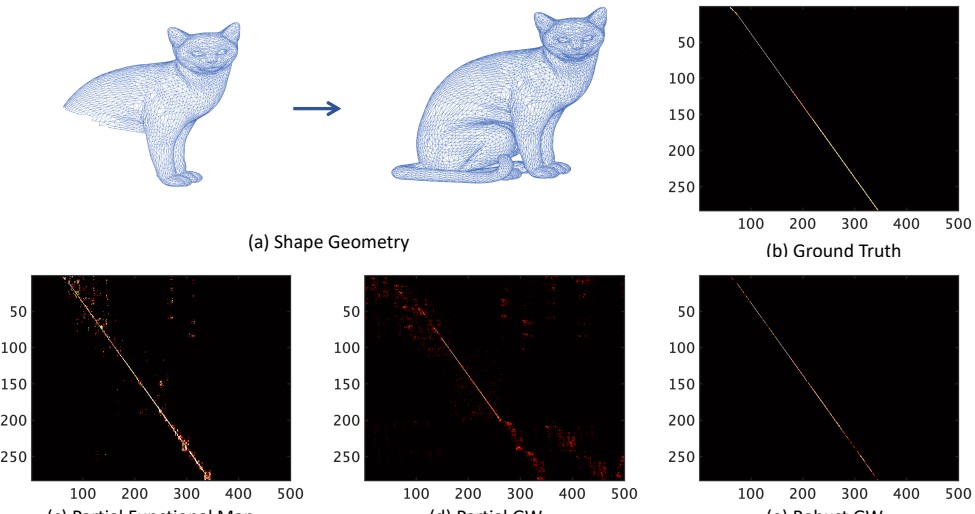

(a) Shape Geometry

(b) Ground Truth

(c) Partial Functional Map

(d) Partial GW

(e) Robust GW

Figure 8: (a): 3D shape geometry of the source and target; (b)-(e): visualization of ground truth, initial point obtained from the partial functional map, and the matching results of PGW and RGW.

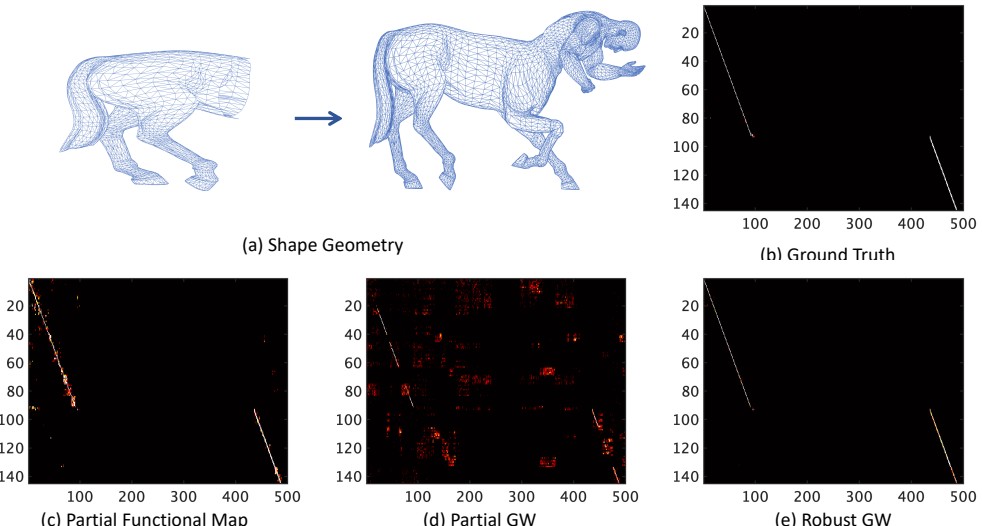

Figure 9: (a): 3D shape geometry of the source and target; (b)-(e): visualization of ground truth, initial point obtained from the partial functional map, and the matching results of PGW and RGW.

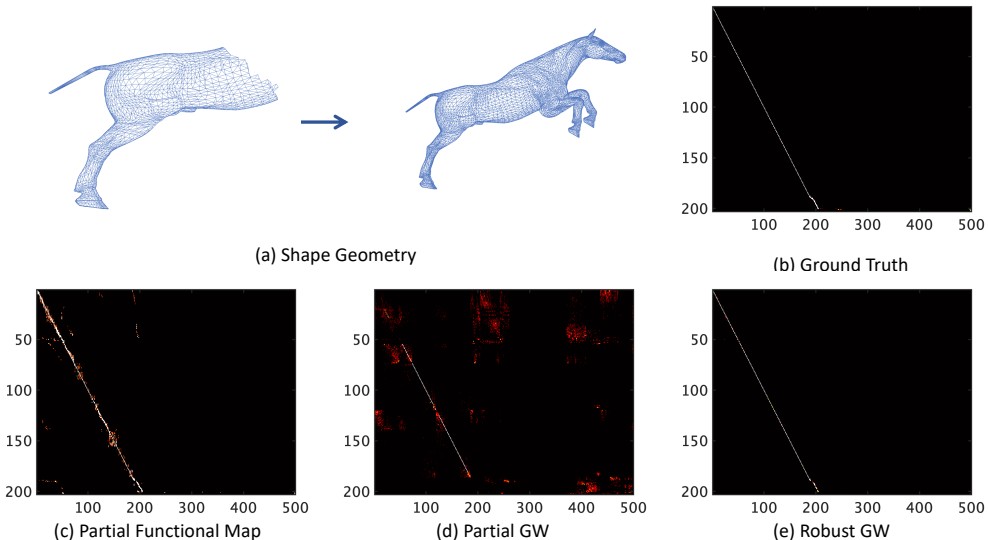

Figure 10: (a): 3D shape geometry of the source and target; (b)-(e): visualization of ground truth, initial point obtained from the partial functional map, and the matching results of PGW and RGW.

