# OpenReview forum: "Outlier-Robust Gromov-Wasserstein for Graph Data"
_NeurIPS.cc/2023/Conference — NeurIPS 2023 spotlight_

### Official Review · Reviewer_YxnT · 2023-06-12

**Soundness:** 3 good
**Presentation:** 3 good
**Contribution:** 3 good
**Rating:** 7
**Confidence:** 4

**Summary:**

This paper proposes a Robust Gromov-Wasserstein (RGW) distance to tackle the high sensitivity to outliers of common GW distances. The key idea is to relax both the marginal distribution and the transport plan based on the KL divergence. An algorithm based on BPALM is further proposed with guaranteed convergence. Theoretical analysis demonstrates the robustness to Huber-$\epsilon$ contamination, and empirical results demonstrate the effectiveness in partial correspondence with the presence of outliers.

**Strengths:**

- The paper is well-written and easy to follow, with theoretically and empirically sounded analysis.
- The paper tackles the crucial robustness problem in GW distances, which could potentially benefit many OT-related research.
- Experiment results demonstrate the effectiveness of RGW on the partial correspondence task, a task that is hard to addressed by most OT methods.


**Weaknesses:**

- A section on related works is **strongly encouraged**. It would be beneficial to discuss the relation/superiority of the proposed RGW compared with other relaxed GW methods, e.g., UGW and srGW.
- Some modifications in the experiment part are encouraged for better presentation. See details in the “Questions” section.


**Questions:**

- RGW imposes an extra relaxation on the marginal distribution, i.e., $KL(\alpha|\mu)<\rho$. Comparing with UGW (UGW is equivalent to $GW_{0,0}$), can you illustrate the benefit/intuition of imposing this extra relaxation?
- For Theorem 2.3, it is possible to have a similar result when both marginals $\mu, \nu$ are contaminated?
- Line 161-162: the statement “By appropriately selecting …  closely approximates the true GW distance (without outliers).” is inaccurate. Setting the second term in Theorem 2.3 only indicates achieving a tighter upper bound but does not necessarily indicates a better approximation.
- For eq. (4-6), why the proximal operator is $KL(\pi|\pi^k)$ in Eq.(4) but $KL(\alpha^k|\alpha)$ in Eq.(5)? Since KL divergence is not symmetric, the authors should justify their choices here.
- Experiments:
  - It is encouraged to show an empirical evaluation on the convergence of the proposed BPALM.
  - A key property of RGW is being robust to outlier. In the subgraph alignment part, will RGW achieve more significant improvements when the query graph is not an exact subgraph, but rather a subgraph with noises (e.g., pseudo nodes/edges) added or deleted?
  - For table 1, why different metrics are used for different datasets?
  - For figure 5, It would be good to show the results with $\rho=0$. i.e., no relaxation on the marginal, to justify the effect of marginal relaxation.
- Typo: the abbreviation for fused Gromov-Wassertein should be FGW instead of FW.


**Limitations:**

The authors properly addressed the limitations in the paper.

---

> ### Author Rebuttal · Authors · 2023-08-09
>
> Thank you for your review and feedback. In the upcoming sections, we will address the questions you've raised.
>
> ---
> **Q1:** A section on related works is strongly encouraged. It would be beneficial to discuss the relation/superiority of the proposed RGW compared with other relaxed GW methods, e.g., UGW and srGW.\
> **Response:** Thanks for your suggestion. We have taken it into account and will include the section mentioned in the response for Q1 in the global response part in our updated version.
>
> ---
> **Q2:** RGW imposes an extra relaxation on the marginal distribution, i.e., $KL(\alpha | \mu) < \rho$. Comparing with UGW (UGW is equivalent to $\textnormal{GW}^{\textnormal{rob}}_{0,0}$), can you illustrate the benefit/intuition of imposing this extra relaxation?\
> **Response:** The extra relaxation in RGW draws inspiration from robust OT [Balaji et al., 2020], optimizing the transport plan and selecting the best marginal distribution within a neighborhood of the given distribution. Perturbed marginal distributions effectively re-weight the samples, reducing the impact of outliers. Introducing this additional relaxation enhances RGW's robustness to the hyperparameters $\rho$ and $\tau$, as demonstrated in Section 4.2. Unlike UGW, which is sensitive to penalty parameters, balancing outlier impact reduction and the control of marginal distortion in the transport plan (lines 44-45), RGW achieves greater flexibility by controlling marginal distortion through $\tau$ and reducing outlier impact using $\rho$.
>
> ---
> **Q3:**  Line 161-162: the statement “By appropriately selecting … closely approximates the true GW distance (without outliers).” is inaccurate. Setting the second term in Theorem 2.3 only indicates achieving a tighter upper bound but does not necessarily indicates a better approximation.\
> **Response:** Thank you for pointing out the inaccuracy in our statement. We will correct this in our revised paper to reflect that the chosen parameters help us obtain a tighter upper bound, rather than a closer approximation to the true GW distance. For a more comprehensive discussion on the tightness of the bound in Theorem 2.3, please refer to our answer to Q3 in the global response section.
>
> ---
> **Q4:**  For Eq. (4-6), why the proximal operator is $KL(\pi | \pi^k)$ in Eq.(4) but $KL(\alpha^k|\alpha)$ in Eq.(5)? Since KL divergence is not symmetric, the authors should justify their choices here.\
> **Response:** The choices facilitate efficient computation in solving the subproblems. Using $KL(\pi | \pi^k)$ in Eq. (4) allows us to directly adopt the Sinkhorn algorithm while employing $KL(\alpha^k|\alpha)$ in Eq. (5) enables a closed-form solution. Importantly, these choices do not affect the convergence results (Theorem 3.3) and provide computational benefits.
>
> ---
> **Q5:** It is encouraged to show an empirical evaluation on the convergence of the proposed BPALM.\
> **Response:** Thanks for your suggestion. We have taken your advice and included plots that illustrate the convergence using different step sizes. You can find these plots in **Figure 3 of the PDF in the global response section**.
>
> ---
> **Q6:** A key property of RGW is being robust to outliers. In the subgraph alignment part, will RGW achieve more significant improvements when the query graph is not an exact subgraph, but rather a subgraph with noises (e.g., pseudo nodes/edges) added or deleted?\
> **Response:** We have conducted an experiment adding 10\% pseudo edges to the subgraphs, and the preliminary findings are in **Table 2 of the global response section's PDF**. These results highlight that RGW significantly outperforms other methods on all the datasets even in the presence of noise. The complete results will be included in our updated paper.
>
> ---
> **Q7:** For table 1, why different metrics are used for different datasets?\
> **Response:**  Hit@1 is equivalent to accuracy, and we use the metric Hit@k for the Douban dataset, which aligns with previous work s for consistency [A, B].\
> [A] Gao, Ji, Xiao Huang, and Jundong Li. "Unsupervised graph alignment with Wasserstein distance discriminator." Proceedings of the 27th ACM SIGKDD Conference on Knowledge Discovery \& Data Mining. 2021.\
> [B] Tang, Jianheng, et al. "Robust attributed graph alignment via joint structure learning and optimal transport." arXiv preprint arXiv:2301.12721 (2023).
>
> ---
> **Q8:** For Figure 5, It would be good to show the results with $\rho=0$. i.e., no relaxation on the marginal, to justify the effect of marginal relaxation.\
> **Response:** Thanks for your suggestion. We have included these results in **Figure 2 of the PDF in the global response section**. The comparison clearly shows that marginal relaxation mitigates the influence of outliers. Specifically, the accuracy of $\rho$ in the range of 0.1 to 1 is significantly higher than when $\rho=0$, justifying the effect of marginal relaxation in our model.
>
> ---
> **Q9:** The abbreviation for fused Gromov-Wasserstein should be FGW instead of FW.\
> **Response:** Apologies for any confusion. In Table 1, the term ``FW'' refers to the Frank-Wolfe method introduced in [Titouan et al., 2019].

---

> > ### Comment · Reviewer_YxnT · 2023-08-12
> >
> > Thank you for the clarifications and additional experiments to answer my concerns, which should be included in the future version. After reading through all the reviews and rebuttal, I would like to increase my score to 7 (accept).

---

### Official Review · Reviewer_Lqoh · 2023-06-19

**Soundness:** 4 excellent
**Presentation:** 4 excellent
**Contribution:** 3 good
**Rating:** 8
**Confidence:** 4

**Summary:**

The authors propose a robust version of the Gromov Wasserstein (RGW) distance and theoretically bound their RGW distance applied to noisy distances in terms of the GW distance between the clean distributions and an error term depending on the choice of parameters in RGW and the distance between the clean distribution and the outlier distribution. The authors develop an algorithm for computing the RGW distance using Sinkhorn’s algorithm and Newton’s method.

**Strengths:**

The theory provides guidance on how to choose parameters in order to get a good approximation (at least an upper bound) of the RGW distance in terms of the GW between clean distributions.

**Weaknesses:**

To get the upper bound (mentioned above) one needs to know the GW distance between the clean distributions.

**Questions:**

(1) How can one estimate the GW distance between clean distribution?
(2) Can Theorem 2.3 be extended to where \nu also has an outlier distribution?
(3) An obvious question, but can one hope to have a lower bound on the RGW distance? Numerically, I would have thought this would be fairly simple to investigate (for example in the context of the examples in Section 4.1) by computing GW of the clean distributions and then varying \rho in the RGW to see how close the upper bound in Theorem 2.3 comes to RGW.

**Limitations:**

The authors close the paper by discussing the heavy computational cost of the method.

---

> ### Author Rebuttal · Authors · 2023-08-09
>
> We appreciate your dedicated review of our manuscript and the valuable feedback you've provided. In the following, we will address the questions and points you've raised in your review.
>
> ---
> **Q1:** To get the upper bound (mentioned above), one needs to know the GW distance between the clean distributions. How can one estimate the GW distance between clean distribution?\
> **Response:** The primary goal of RGW is to estimate the GW distance between clean distributions in the presence of outliers. Theorem 2.3 establishes that RGW provides a lower bound on the true GW distance between clean distributions and offers guidance on parameter selection for obtaining a tighter lower bound of the GW distance without outliers. This indicates RGW's ability to estimate the GW distance between clean distributions to some extent. BPALM can be efficiently utilized to compute RGW, enhancing its practicality for real-world applications.
>
> ---
> **Q2:** Can Theorem 2.3 be extended to where $\nu$ also has an outlier distribution?\
> **Response:** Yes. Please refer to the response provided for Q2 in the global response.
>
> ----
> **Q3:** An obvious question, but can one hope to have a lower bound on the RGW distance? Numerically, I would have thought this would be fairly simple to investigate (for example in the context of the examples in Section 4.1) by computing GW of the clean distributions and then varying $\rho$ in the RGW to see how close the upper bound in Theorem 2.3 comes to RGW.\
> **Response:** For general scenarios, where the GW distance between clean samples is not necessarily zero, the construction of a lower bound represents a non-trivial challenge. This difficulty stems from the nonconvexity of RGW. Notably, even in the case of robust OT [Balaji et al., 2020], which is a convex problem, only an upper bound has been established.  Following your suggestion, we conducted an experiment to analyze how the function value of RGW and its upper bound changes when varying $\rho$ of the toy example in Section 4.1. In **Figure 1(b) in the PDF in the global response section**, the results demonstrate that the function value of RGW and the upper bound converge to zero as $\rho$ increases, which aligns with the theoretical perspectives presented in Theorem 2.3. While the building of a precise lower bound for RGW remains a complex task, it will be a direction for our future research. Please refer to the response for Q3 in the global response for further discussion on the tightness of the bound in Theorem 2.3 and the lower bound.

---

> > ### Comment · Reviewer_Lqoh · 2023-08-21
> >
> > Thank you for your answers: I'm happy with the response and I appreciate you performing extra experiments. I increased my score.

---

### Official Review · Reviewer_1vP9 · 2023-07-06

**Soundness:** 2 fair
**Presentation:** 3 good
**Contribution:** 3 good
**Rating:** 7
**Confidence:** 5

**Summary:**

The authors propose a new formulation of unbalanced Gromov-Wasserstein (GW), called Robust Gromov-Wasserstein (RGW). Instead of controlling the deviations of the OT marginals from the prior marginal distributions $\mu$ and $\nu$ directly via quadratic phi-divergences like UGW, RGW proposes to use linear phi-divergences (KL) operating on the intermediate learned marginals $\alpha$ and $\beta$ which are constrained to be relatively close to $\mu$ and $\nu$ w.r.t the KL. It is shown that this new formulation is theoretically better at handling outliers than the UGW under the assumption of Huber's contamination model.

They then suggest using a Bregman proximal alternating linearisation minimisation (BPALM) algorithm to solve this problem, which is shown to converge asymptotically to a critical point. Finally, the authors show empirically on a synthetic dataset that their method is more robust to outliers, and that it leads to better performance than many competitors on subgraph alignments and partial shape correspondences.

**Strengths:**

-	Overall I find the paper well-written and I didn’t find any mistake in the proofs.
-	RGW is novel in the Gromov-Wasserstein literature, even if inspired from prior work on unbalanced linear OT as mentioned by authors. The theoretical results on robustness are compeling. The solver proposed by authors seem clearly competitive compared to the one for UGW, while its use is also theoretically supported.
-	RGW shows better results than competitors on partial shape matching of both synthetic and real-world data.
-	RGW also shows better results than competitors on subgraph alignment problems on many real-world graph datasets. Authors empirical study considers subgraphs of various proportions sampled from given input (full) graphs. RGW shows consistent improvements compared to competitors along these various proportions. Moreover authors show that RGW performances are rather consistent within considered sets of validated hyperparameters.


**Weaknesses:**

 - 1. Positioning w.r.t the unbalanced OT literature:
     - a) *[addressed by authors]* The extension to GW of the work on Wasserstein in [Balaji & al, 2020] should be more explicit. As the latter does not consider regularizing their loss function, you should justify why you do so in your GW setting.

     - b) *[addressed by authors]* The recent paper of [A] provides in a sense a primal point of view of robust GW, It would be good to discuss it and to benchmark it in your subgraph alignment benchmark.

- 2. Robustness guarantees :
     - a) *[addressed by authors]* Missing proof: Could you provide the proof for UGW in Remark 2.4 ?

     - b) *[addressed by authors]* Tightness: Could you empirically analyze the tightness of both bounds for instance while e.g validating algorithm initializations to get better estimates of various GW terms ?

     - c) *[addressed by authors]* Positioning with recent analysis: Theorem 1 in [Tran & al, 2023] also provides an affine relation of the form: noisy UCOOT <= a * clean UCOOT + b. I believe that this theorem can be easily extended to UGW so could you discuss the relation of this theorem to your bounds ?

- 3. Benchmark:
     - a) *[addressed by authors]* RGW depends on a lot of hyperparameters. Authors show that their performances were rather consistent for the hyperparameters tau and rho which are involved in the RGW program however it seems that the sensitivity analysis always includes an intensive validation of the BPALM step sizes hyperparameters.  i) Could you clarify if shown results in Section 4.2 always includes this validation of BPALM hyperparameters ? ii) Could you provide a sensitivity analysis to these hyperparameters ?

     - b) *[partially addressed by authors using normalized degrees]* Sensitivity to input graph distribution:  In the context of graph representation learning, an open question relates to a good choice of input graph distribution for tasks like graph partitioning [Xu & al, 2019a ; Chowdhury & al, 2021; Vincent-Cuaz & al , 2022]. Uniform distribution is the common simple choice which seems to be a good enough choice, but power-law distributions based on normalized degrees clearly allow better partitioning performances. Could you integrate these kind of considerations in your benchmark for subgraph alignment ? That would also provide insights on the robustness of RGW to these hyperparameters inherent to all transport problems.


[A] Liu, Weijie, et al. "Robust Graph Dictionary Learning." The Eleventh International Conference on Learning Representations. 2022.

NB: the citation model seems to be the wrong one for Neurips, currently names and dates instead of simple numbers.

*Update after rebuttal : The authors have addressed most of my concerns through their rebuttals and discussions. Considering that the authors have undertaken to amend the paper and supplementary material accordingly, I increase my grade from 5 (borderline accept) to 7 (accept).*

**Questions:**

I refer the authors to the weaknesses' list above for suggestions and questions that I encourage them to discuss. A last suggestion:
- I believe both discussions and proofs for the Newton algorithm could be simplified simply observing that $p$ is strictly convex on the positive orthant.

**Limitations:**

I think discussing the weaknesses I've mentioned and answering my questions detailed above would help identify and work around the other limitations of their current work. If the authors manage to address these, I will gladly increase my rating.
This work has no negative societal impact.

---

> ### Author Rebuttal · Authors · 2023-08-09
>
> We appreciate your thoughtful review and the constructive feedback you have provided.  In what follows, we answer the following questions pointed out in your review.
>
> ---
> **Q1:** The extension to GW of the work on Wasserstein in [Balaji et al, 2020] should be more explicit. As the latter does not consider regularizing their loss function, you should justify why you do so in your GW setting.\
> **Response:** Please refer to the response provided for Q1 in the global response section. We will include more explanations about why we relax the hard marginal constraints in the updated paper.
>
> ---
> **Q2:** The recent paper of [A] provides in a sense a primal point of view of robust GW, It would be good to discuss it and to benchmark it in your subgraph alignment benchmark.\
> **Response:** We would like to acknowledge that the problem addressed in [A] is focused on tackling the inaccuracy of pairwise relation matrices (PRM), the matrices $D$ and $\bar{D}$ in Eq. (3). The authors propose an approach by introducing a perturbation matrix to allow PRM to vary in a neighborhood of the observed PRM, seeking a transport plan that minimizes transportation costs for all perturbations. While their idea differs from ours, we recognize its potential to help reduce the impact of outliers by perturbing the distance between clean samples and outliers. In our paper, we plan to include RGWD introduced in [A] as one of the baselines in the subgraph alignment task. We have already provided preliminary results of RGWD on the 50\% subgraph alignment in the following table, and we plan to present the complete results in the updated paper.
>
> Table: The performance of RGWD of 50\% subgraph on datasets Synthetic, Proteins and Enzymes and Douban.
>
> |     | Synthetic |     | Proteins-1 |     | Proteins-2 |     | Enzymes |     | Douban |     |
> | --- | --- | --- | --- | --- | --- | --- | --- | --- | --- | --- |
> | Method | Acc | Time | Acc | Time | Acc | Time | Acc | Time | Hit@1 | Hit@10 |
> | RGWD | 16.68 | 955.40 | 27.94 | 4396 | 59.69 | 3586 | 30.35 | 2629 | 4.11 | 16.46 |
> | Our RGW | 94.44 | 361.44 | 53.30 | 834.76 | 69.38 | 466.91 | 63.43 | 293.84 | 75.58 | 96.24 |
>
> ---
> **Q3:** Missing proof: Could you provide the proof for UGW in Remark 2.4?\
> **Response:** In Remark 2.4, the case where only $\mu$ is corrupted is considered. Given that $\pi_c$ is a feasible solution to the UGW problem, plugging it in yields $d_{\textbf{KL}}^\otimes (\nu,\nu)=0$, leading to the desired result.
>
> ---
> **Q4:** Tightness: Could you empirically analyze the tightness of both bounds for instance while e.g validating algorithm initializations to get better estimates of various GW terms?\
> **Response:** We have conducted experiments to investigate the tightness when varying the relaxation parameter $\rho$ and penalty parameter $\tau$. For a detailed explanation and insights into these experiments, please refer to our response to Q3 in the global response section.
>
> ---
> **Q5:** Positioning with recent analysis: Theorem 1 in [Tran et al., 2023] also provides an affine relation of the form: noisy UCOOT <= a * clean UCOOT + b. I believe that this theorem can be easily extended to UGW so could you discuss the relation of this theorem to your bounds?\
> **Response:** Thanks for pointing out this issue. We would like to emphasize that the baseline used in [Tran et al., 2023] is UGW but the baseline used in Remark 2.4 is GW. Specifically, the inequality constructed in [Tran et al., 2023] on the right-hand side is an affine function of UGW, whereas, in Remark 2.4, the right-hand side of the inequality is an affine function of GW. The result in [Tran et al., 2023] doesn't directly correlate UGW and GW as mentioned briefly in lines 42-44. We address this by establishing the UGW and GW relationship in Remark 2.4.  Theorem 2.3 establishes that RGW is robust to outliers and further establishes a clear relationship between RGW and GW. We will incorporate your suggestions to provide a more detailed exploration of this point in our discussion.
>
> ---
> **Q6:** However it seems that the sensitivity analysis always includes an intensive validation of the BPALM step sizes hyperparameters. i) Could you clarify if shown results in Section 4.2 always includes this validation of BPALM hyperparameters? ii) Could you provide a sensitivity analysis to these hyperparameters?\
> **Response:** In the subgraph alignment task, we have used constant values for the stepsizes $t_k$, $c_k$ and $r_k$. We have conducted a sensitivity analysis for these parameters, with the findings detailed in **Tables 3 and 4 in the PDF of the global response section**. Specifically, **Table 3** reveals that RGW achieves its highest accuracy with $t$ in the range of 0.01 to 0.05, allowing us to select $t=0.01$ as the default. **Table 4** further indicates that accuracy is not significantly affected by variations in $c$, leading us to set $c=0.1$ as the default.
>
> ---
> **Q7:** Sensitivity to input graph distribution: Power-law distributions based on normalized degrees clearly allow better partitioning performances. Could you integrate these kind of considerations in your benchmark for subgraph alignment? That would also provide insights on the robustness of RGW to these hyperparameters inherent to all transport problems.\
> **Response:** Thank you for your suggestion. In **Table 1 of the PDF in the global response section**, we present the results of using the normalized degree (a special case of power-law distribution) as the input graph distribution for the 50\% subgraph alignment task. The findings affirm that our proposed RGW outperforms other methods in accuracy when using normalized degrees as marginal distributions. The complete results will be included in our updated paper.

---

> > ### Comment · Reviewer_1vP9 · 2023-08-10
> > **Reply to authors**
> >
> > Thank you for your answers, overall I found authors' rebuttal convincing and will increase my grade. Some remarks and questions:
> >
> > - All complementary results provided in the rebuttal pdf and other answers are good and of interest thus should be mentioned in the paper and reported in the supplementary material with adequate experimental details (RGWD should be included in the main paper). Could you please detail the experiments you did for Figure 1 in the pdf ? it would be of interest to observe mean and std for different noise samples.
> >
> > - **Q5**: from my understanding, as $UGW \leq GW$, once that one established that noisy UGW $\leq$ clean UGW (up to affine transform following [Tran & al 2023]), it follows that noisy UGW $\leq$ clean GW (up to affine transform). This way we can envision that UGW also gives an estimation of clean GW but Remark 2.4 provides a kind of impossibility result in practice. So indeed it would be good to emphasize this relation with the Theorem from [Tran & al 2023] (plus potentially Theorem 2 in UGW's paper on the tightness of bi-convex relaxation).

---

> > > ### Author Response · Authors · 2023-08-11
> > > **Response to Reviewer 1vP9**
> > >
> > > Thank you for your kind response and appreciation of our contributions. We will answer your follow-up questions in the following.
> > > - For the setting of Figure 1 in the PDF, we follow the setting of the toy example in Section 4.1. The original shape is created by
> > > $$(x,y) = (\cos(z)+0.1\sin(z),0.5\sin^3(2z),0.5\sin^3(2z)+0.1\cos(5z),~z\in (0,2\pi).$$
> > > We sampled 300 points from the source shape and 400 points from the target shape. We introduced outliers by randomly adding points from a discrete uniform distribution on $[-3,-2.5]\times [0,0.5]$. In Figure 1(a) in the PDF in the global response section, we conducted experiments for the outlier ratio $\epsilon$ varies from 0 to 0.5. For Figure 1(b) in the PDF in the global response section, we fixed the number of outlier to be 50 ($\epsilon$ = 1/7), which is the same setting in Figure 2 in the paper, and varied the value of relaxation parameter $\rho$ to see how the value of RGW and its upper bound change. Besides, for Figure 1(c) in the PDF in the global response section, we used the same approach to generate data as in Figure 1(b) and we altered the value of penalty parameter $\tau$ to observe the change of the value of UGW and its upper bound. Following your suggestion, we added Gaussian noise to 10\% of the point of the clean sample in source domain. Since we are not allowed to modify the PDF in this period, we present the mean and std of different methods in 5 trials over different random seeds in the noise generating process in the following tables. Table 1, 2, 3 corresponds to Figure 1 (a), (b), (c) in the PDF in the global response respectively. We will include all the results, including the complete results of RGWD and all the experiments reported in the PDF in the global response in our updated paper as we will have an extra page after the acceptance.
> > >
> > > Table 1: Function values (Mean $\pm$ Std.) of UGW, PGW, and RGW for varying $\epsilon$.
> > >
> > > | Method | $\epsilon=0$ | $\epsilon=0.1$ | $\epsilon=0.2$ | $\epsilon=0.3$ | $\epsilon=0.4$ | $\epsilon=0.5$ |
> > > | --- | --- | --- | --- | --- | --- | --- |
> > > | UGW | $9.00\times10^{-3}\pm 1.66\times 10^{-4}$ | $0.026\pm 0.0028$ | $0.044\pm 0.0001$ | $0.053\pm 0.0001$ | $0.062 \pm 0.0019$ | $0.071\pm 0.0002$ |
> > > | PGW | $7.29\times 10^{-5} \pm 5.50\times 10^{-6}$ | $0.025\pm 0.0023$ | $0.084\pm 0.0005$ | $0.148\pm 0.0009$ | $0.183 \pm 0.0012$ | $0.214\pm 0.0006$ |
> > > | RGW | $5.05\times 10^{-4}\pm 8.55\times 10^{-5}$ | $0.013\pm 0.0001$ | $0.014\pm 0.0031$ | $0.026 \pm 0.0001$ | $0.033 \pm 0.0001$ | $0.037 \pm 0.0001$ |
> > >
> > > Table 2: Function value (Mean $\pm$ Std.) of RGW and its upper bound for different $\rho$
> > >
> > > |     | $\rho=0.01$ | $\rho=0.1$ | $\rho=1.0$ | $\rho=5.0$ |
> > > | --- | --- | --- | --- | --- |
> > > | RGW-UB | $0.116\pm 0.0001$ | $0.108\pm 0.0001$ | $0.034\pm 0.0001$ | $9.77\times 10^{-4}\pm 0.0001$ |
> > > | RGW | $0.018\pm 0.0001$ | $0.014\pm 0.0002$ | $0.011\pm 0.0009$ | $1.54\times 10^{-5}\pm 1.68\times 10^{-6}$ |
> > >
> > > Table 3: Function value (Mean $\pm$ Std.) of UGW and its upper bound for different $\tau$
> > >
> > >   |     | $\tau=0.01$ | $\tau=0.1$ | $\tau=1.0$ | $\tau=2.0$ |
> > >   | --- | --- | --- | --- | --- |
> > >   | UGW-UB | $0.014\pm 0.0001$ | $0.135\pm0.0001$ | $1.346\pm 0.0001$ | $2.692\pm 0.0001$ |
> > >   | UGW | $0.012\pm 9.82\times 10^{-5}$ | $0.045\pm 0.0023$ | $0.280\pm 0.0003$ | $0.400\pm 0.0005$ |
> > >
> > > - Thanks for pointing this out. The observation about the relation between the UGW and GW you mentioned is insightful and we will emphasize this relation with Theorem 1 from [Tran \& al 2023] and Theorem 2 in UGW's paper in the updated paper.

---

> > > > ### Author Response · Authors · 2023-08-15
> > > >
> > > > Please let us know if our response addresses your concerns. We are happy to address any remaining points during the discussion phase.

---

> > > > > ### Comment · Reviewer_1vP9 · 2023-08-15
> > > > > **Reply to authors 2**
> > > > >
> > > > > Thank you for these answers. You addressed all of my concerns so I updated my review and increased my rating towards acceptance from 5 to 7.

---

### Official Review · Reviewer_we1a · 2023-07-07

**Soundness:** 2 fair
**Presentation:** 3 good
**Contribution:** 2 fair
**Rating:** 4
**Confidence:** 4

**Summary:**

The authors propose a new variant for unbalanced Gromov-Wasserstein with Kullback-Leibler (KL) divergence for marginal relaxation (Sejourne et al., 2021) by leveraging the outlier-robust approach (Mukherjee et al., 2021). The authors propose an algorithm approach by using Bregman proximal alternating linearization. The authors illustrate the advantages of the proposed method on several experiments.

**Strengths:**

+ The authors propose a new variant for Unbalanced GW with KL divergence for marginal relaxation (Sejourne et al., 2021) by extending the outlier-robust approach (Mukherjee et al., 2021).
+ The authors propose to use the Bregman Proximal Alternating Linearized Method to optimize the proposed outlier-robust GW.
+ The authors empirically demonstrate the advantages of the proposed approach.

**Weaknesses:**

+ The authors leverage the outlier-robust approach (Mukherjee et al., 2021) for unbalanced GW. The proposed method is essentially a new variant of unbalanced GW, but somewhat expected results.
+ Some claims and experimental results are needed to elaborate with more details (see the Questions part)

**Questions:**

The ideas of the proposed method are clear. It is easy to follow the presentation. However, some parts are needed to elaborate with more details to clarify the claims and the contributions.
+ The authors should discuss the relation between the proposed outlier-robust GW with the approach in (Sejourne et al., 2021) for unbalanced GW and (Mukherjee et al., 2021) for outlier-robust approach.
+ As in Definition 2.2, the authors propose to use KL-divergence instead of quadratic KL-divergence (as in Sejourne et al., 2022) due to non-convexity. However, the problem (1) is non-convex. It is not clear the advantages of using KL-divergence over quadratic KL-divergence. Could the authors elaborate more details on this point? Additionally, as  results in Theorem 3.3, could the authors comment the gain of such choice?
+ The quadratic KL-divergence helps to maintain the homogeneity of GW while KL-divergence will result in non-homogeneous unbalanced GW (especially when input measures have small or large total mass). Could the authors comment on the choice of KL-divergence over the quadratic KL-divergence?
+ For Theorem 2.3, is the bound tight? Additionally, Theorem 2.3 only address one of the input measures is corrupted, but not two input measures may be corrupted as in problem (1). Is there any limitation for this result? It is better if the authors extend this result for the case as considered in problem (1).
+ As in line 161-162, Theorem 2.3 gives the upper bound only, it is not clear why “robust GW value that closely approximates the true GW distance (without outliers)”.
+ It seems that the Proposition 3.2 is a standard optimization result about the unique solution for the root of a monotonic function?
+ For results in Figures 2 and 3, how to choose the hyperparameters? Do the hyperparameters affect the results?
+ Could the authors clarify the stopping condition of algorithms for the reported time? (since the problem is non-convex, are any algorithms sensitive to the initialization?)
+ As in line 285-288, could the authors elaborate how to choose the best result for those hyperparameters?
+ For results in Figure 5, it is surprising that the hyperparameters have negligible effects on accuracy when one maintains some fixed ratio? However, it is not clear how one can choose a suitable ratio? I suggest that it seems better to extend the range of $\rho$ and $\tau$ for results in Figure 5. Could the authors explain results in Figure 5 with more details?

Minor point:
+ It is better to add the standard deviation for results in Table 1, Figure 4?

---

Thank you for the rebuttal.

**Limitations:**

The authors have discussed limitations of their work. However, there is no discussion about the potential negative societal impact of their work.

---

> ### Author Rebuttal · Authors · 2023-08-09
>
> Thank you for your instructive comments. We would like to emphasize that RGW  definitely goes beyond the mere combination of outlier robust OT and unbalanced GW. Our primary contribution lies in the meticulous design of an objective function that offers the advantages of both enhanced model efficiency and computational effectiveness.
> * **Effective Model and Statistical Guarantee:**  Firstly, instead of utilizing the quadratic KL divergence as in unbalanced GW, we leverage the convexity of KL divergence to establish the statistical properties of RGW, leading to an upper bound. This upper bound is controllable through the hyperparameters $\tau$ and $\rho$. It serves as a guiding principle to assist us in effectively tuning parameters in practice. Notably, this favorable property ceases to hold for quadratic KL divergence due to its nonconvex nature.
> * **Computational Efficiency:** Secondly, employing KL divergence allows us to solve the subproblems in a highly efficient manner, as we detail in the response to Q2 below. Furthermore,  our RGW formulation is a nonconvex problem, which presents fundamental challenges for algorithm design compared with OT-related problems. To overcome this, we develop an efficient algorithm called BPALM  tailored to RGW.
> * **Exceptional Numerical Results:** Our confidence in the superiority of RGW stems from comprehensive numerical results that highlight its superior performance compared to other methods.
>
> Below we provide a detailed response to your questions. We believe that our responses will address your concerns, leading to a reevaluation of our paper's contribution and quality.
>
> ---
> **Q1:**  The authors should discuss the relation between the proposed outlier-robust GW with the approach for unbalanced GW for outlier-robust approach.\
> **Response:**  Please refer to the response provided for Q1 in the global response section. We will include a related work section in the updated version.
>
> ---
> **Q2:** Could the authors comment on the choice of KL-divergence over the quadratic KL-divergence?\
> **Response:** First, we chose KL-divergence over quadratic KL-divergence for its computational advantages, allowing convex subproblems with closed-form solutions, unlike the nonconvex quadratic KL-divergence which requires linearization for the subproblems in each iteration. Though linearization increases computational complexity due to the lack of closed-form solutions, it still allows for convergence analysis (like Thm 3.3). Second, KL-divergence provides an upper bound with explicit control through hyperparameters, offering a statistical advantage that would be disrupted by the nonconvex nature of quadratic KL-divergence. Besides, rather than maintain homogeneity, the KL-divergence aligns with our main objective of eliminating outliers and is less sensitive to them, unlike the more outlier-sensitive quadratic KL-divergence.
>
> ---
> **Q3:** For Theorem 2.3, is the bound tight? It is better if the authors extend this result for the case as considered in problem (1).\
> **Response:** For the tightness of the bound in Theorem 2.3, please refer to the responses for Q3 in the global response section. Besides, Theorem 2.3 can be extended to scenarios where both input measures are corrupted. For a thorough discussion of this extension, kindly refer to the responses for Q2 in the global response section.
>
> ---
> **Q4:** It seems that Proposition 3.2 is a standard optimization result about the unique solution for the root of a monotonic function?\
> **Response:** Yes. Most of Newton's method convergence results focus on local convergence. We establish global convergence for convex monotonic functions for completeness.
>
> ---
> **Q5:** For results in Figures 2 and 3, how to choose the hyperparameters? Do the hyperparameters affect the results? \
> **Response:** The choice of hyperparameters in Figures 2 and 3 is obtained using the method discussed in section 4.2, where the hyperparameters affect the outcomes, as highlighted in the analysis in section 4.2 and illustrated in Figure 5.
>
> ---
> **Q6:** Could the authors clarify the stopping condition of algorithms for the reported time? (since the problem is non-convex, are any algorithms sensitive to the initialization?)\
> **Response:** The stopping criterion is determined by $\max_{i,j} |\pi_{ij}-\pi^{prev}_{ij}|$, which is the same as used in UGW. All methods in section 4.2 are sensitive to initialization. We use the same initialization for evaluating their performance stated in lines 289-292.
>
> ---
> **Q7:** As in line 285-288, could the authors elaborate on how to choose the best result for those hyperparameters?\
> **Response:**  We conduct experiments with various combinations of $\rho$, $\tau$, and stepsizes within the range specified in lines 285-288 and report the highest accuracy achieved.
>
> ---
> **Q8:** Question for Figure 5. It is not clear how one can choose a suitable ratio? I suggest that it seems better to extend the range of $\rho$ and $\tau$ for results in Figure 5. Could the authors explain Figure 5 with more details?\
> **Response:** In Figure 5, the term 'ratio' refers to the subgraph's node proportion to the entire graph, a value dictated by the dataset. We have extended the range of $\rho$ and $\tau$, with results in **Figure 2 in the PDF of the global response section**. The accuracy declines when $\rho$ is too large or too small but is robust between 0.05 and 1, leading us to set $\rho=0.2$. A similar pattern was observed for $\tau$ leading us to set $\tau=0.1$ as the default.
>
> ---
> **Q9**: It is better to add the standard deviation for results in Table 1, Figure 4? \
> **Response:** We have conducted an experiment adding 10\% pseudo edges to the subgraphs, and the preliminary findings, including the mean accuracy and standard deviation over 5 independent trials, are presented in **Table 2 of the global response section's PDF**. We will include the complete results with standard deviation in the updated paper.

---

> > ### Author Response · Authors · 2023-08-16
> > **Follow-up**
> >
> > We would like to follow up to see if our response addresses your concerns or if you have further questions. We are happy to address any remaining points during the discussion phase.

---

> > ### Comment · Reviewer_we1a · 2023-08-18
> >
> > Thank you for the rebuttal. I have no other raised points.

---

### Author Rebuttal · Authors · 2023-08-09

Thanks for all the insightful feedback. We summarize the common questions and provide corresponding responses in this global response section. We'd rather happy to take any other questions during the discussion period.

---
**Q1:** A section focusing on related work should be included to explore and compare the proposed RGW method with other relaxed GW approaches and robust OT techniques.\
**Response:** Thanks for your suggestions. Our introduction of relaxed distributions to handle outliers draws inspiration from robust OT techniques [Balaji et al., 2020]. Unlike robust OT, which is a convex optimal transport problem, RGW is non-convex, presenting algorithmic challenges. RGW replaces hard marginal constraints with a penalty function, inspired by UGW [Sejourne et al., 2021], to mitigate outlier impact. Notably, RGW offers improved robustness and efficiency compared to UGW, which is sensitive to penalty parameters and lacks an efficient algorithm as discussed in lines 44-48. Moreover, srGW corresponds to an extreme case of UGW, achieved by setting one of the penalty parameters to infinity and the other to 0. For a detailed discussion, we plan to include a paragraph titled ``Related Work'' for a comprehensive discussion of robust OT and the relationship between RGW and relaxed GW methods in the updated version.

---
**Q2:** Can Theorem 2.3 be extended to the case where both $\mu$ and $\nu$ are contaminated with outliers? \
**Response:**  Yes. Consider $\mu = (1-\epsilon_1)\mu_c + \epsilon_1 \mu_a$ and $\nu = (1-\epsilon_2)\nu_c + \epsilon_2\nu_a$. Utilizing the same notations and techniques as in the proof of Theorem 2.3, we obtain the following inequality by plugging in $\pi_ c$: \
$\text{GW}^{\text{rob}}_{\rho_1,\rho_2} (\mu, \nu)\leq\inf _{\alpha\in\mathcal{P}(X), \ \beta\in \mathcal{P}(Y)} \text{GW}(\mu_c, \nu) + \tau_1 d _{\textbf{KL}}(\mu _c,\alpha) + \tau_2 d _{\textbf{KL}}(\nu _c,\beta) $ \
$ \qquad\qquad\qquad\qquad\text{s.t.} \\ d _{\textbf{KL}}(\mu,\alpha) \leq \rho_1,~d _{\textbf{KL}}(\nu,\beta) \leq \rho_2.$\
Subsequently, employing the same technique as in the proof of Theorem 2.3 to minimize the right-hand side, we obtain \
$\text{GW} _{\rho_1,\rho_2}^{\text{rob}} (\mu, \nu) \leq \text{GW}(\mu_c, \nu_c)+\max \left(0,\epsilon_1-\frac{\rho_1}{d _{\textbf{KL}}(\mu_a,\mu_c)}\right)\tau_1d _{\textbf{KL}}(\mu_c,\mu_a)+\max \left(0,\epsilon_2-\frac{\rho_2}{d _{\textbf{KL}}(\nu_a,\nu_c)}\right)\tau_2d _{\textbf{KL}}(\nu_c,\nu_a)$.

---
**Q3:** Tightness of the upper bound in Theorem 2.3 and lower bound of RGW. \
**Response:** Thank you for raising this insightful question regarding the tightness of the upper bound in Theorem 2.3 and the lower bound of RGW.
*  **Tightness of Upper Bound:** Since our primary focus is on cases where the GW distance between clean samples is zero or near zero (due to noise), such as in the partial shape correspondence and subgraph alignment tasks, there can be an isometric mapping from the query subgraph to part of the entire graph. By choosing an appropriate value for $\rho$ (as discussed in lines 160-161), the upper bound becomes the GW distance between clean samples, which is zero. As RGW is always nonnegative, the upper bound is tight in this context.
* **Lower Bound:** Addressing the lower bound for general cases, where the GW distance between clean samples is not necessarily zero, presents a non-trivial challenge due to the nonconvexity of RGW. Even robust OT [Balaji et al., 2020], a convex problem, only provides an upper bound. Building the lower bound is complex and will be considered as a potential direction for future research.
* **Empirical Insights:** We have empirically explored the function values of PGW, UGW, and RGW with varying outlier ratios in a toy example (Section 4.1), as shown in **Figure 1(a) in the PDF in the global response section**. The findings confirm that RGW's value can remain close to zero as the ratio of outliers increases. Moreover, **Figure 1(b) in the PDF** depicts the function value of RGW and its upper bound as $\rho$ varies. The function value of RGW and its upper bound decrease, converging to zero as $\rho$ increases. This observation provides a valid corroboration of Theorem 2.3. Furthermore, We have also examined how UGW and its upper bound change with $\tau$ as shown in **Figure 1(c) in the PDF**, noting that the value of UGW and its upper bound increase with enlarging $\tau$.  Unlike RGW, UGW's $\tau$ must balance between the reduction of outlier impact and the preservation of marginal distortion in the transport plan. This demands a careful balance and cautions against setting $\tau$ excessively close to zero, which could lead to over-relaxation.

---

### Decision · Program_Chairs · 2023-09-21

**Decision:**

Accept (spotlight)

**Comment:**

The authors proposed a novel robust GW with a computationally efficient solver. Reviewers mostly appreciated the paper but had a few concerns in their reviews.

The authors did a fairly convincing and detailed rebuttal for most of the questions and provided several new experimental results in the rebuttal PDF. This was much appreciated by the reviewers and two of them increased their score.

For these reasons I propose to accept the paper but strongly suggest that the authors take into account the comments form the reviewers and integrate the new results and discussion in the main paper and supplementary since it clearly makes the paper stronger.